# The unfolded protein response in progeria arteries originates from non-endothelial cell types

Raquel A Silva[1,2,3] , Fatih Sarigol[1,2] , G Elif Karagöz[1,2], Selma Osmanagic-Myers[4] , Roland Foisner[1,2]

**Hutchinson-Gilford Progeria Syndrome (HGPS) is a premature aging disease caused by a mutation in *LMNA*, leading to the expression of a prelamin A variant called progerin. HGPS hallmarks include accelerated cardiovascular disease and atherosclerosis, caused in part by ER stress-induced apoptosis of vascular smooth muscle cells. As a dysregulated unfolded protein response (UPR) can induce endothelial cell (EC) pathology during aging, we investigated whether loss of proteostasis contributes to EC dysfunction in HGPS, using an endothelium-specific HGPS mouse model. Contrary to previous reports in vascular smooth muscle cells and fibroblasts, we found no robust activation of UPR in ECs constitutively expressing progerin, and cells retained the ability to elicit potent UPR when exposed to external ER stress. Unlike aortic tissue derived from mice with endothelium-specific progerin expression, aorta from *Lmna^{G609G/+}* mice with ubiquitous progerin expression showed up-regulation of the UPR, suggesting that the UPR in HGPS aorta is primarily rooted in non-ECs. Analysis of scRNA-Seq datasets from aorta in *Lmna^{G609G/G609G}* mice confirmed this hypothesis. Our data indicate that UPR activation is a cell-type-specific phenomenon in progerin-expressing arteries.**

## Introduction

Hutchinson-Gilford progeria syndrome (HGPS) is a rare, premature aging disease that affects 1 in 20 million people worldwide (https://rarediseases.org/rare-diseases/hutchinson-gilford-progeria/; https://www.progeriaresearch.org/quick-facts/). It is caused by a dominant de novo (c.1824C>T) mutation in exon 11 of the *LMNA* gene (De Sandre-Giovannoli et al, 2003; Eriksson et al, 2003), which encodes for lamins A and C. WT lamin A is expressed as prelamin A and processed post-translationally by several consecutive reactions at the C-terminus to generate mature lamin A. These include farnesylation at the cysteine residue in the

C-terminal CSIM sequence (Fisher et al, 1986; Holtz et al, 1989), cleavage of the last three C-terminal residues, carboxymethylation, and a final cleavage of the last 15 amino acids by the Zmpste24 peptidase, including the farnesyl and carboxymethyl groups (Naetar et al, 2017). The HGPS-linked single base substitution in *LMNA* activates a cryptic splice site that produces a truncated form of prelamin A, which is initially processed like WT prelamin A. However, due to the lack of the recognition sequence for the Zmpste24 peptidase, the final proteolytic cleavage cannot be performed, resulting in a permanently farnesylated and carboxymethylated toxic prelamin A variant, called progerin (Gonzalo et al, 2017). Lamins make up the nuclear lamina, a filamentous protein meshwork that provides structure to the nucleus and plays a crucial role in chromatin organization and gene regulation by interacting with heterochromatin (Gruenbaum & Foisner, 2015). Accordingly, at the cellular level, progerin expression has been linked to nuclear envelope abnormalities, loss of peripheral heterochromatin, gene expression changes, DNA damage accumulation, telomere shortening, and premature cellular senescence (Foo et al, 2019; Cenni et al, 2020; Primmer et al, 2022).

HGPS patients present with growth retardation, alopecia, skin alterations, lipodystrophy, and osteolysis (DeBusk, 1972; Hennekam, 2006; Merideth et al, 2008; Batista et al, 2023). One of the hallmarks of HGPS is accelerated cardiovascular disease (CVD) with associated severe atherosclerosis (Olive et al, 2010; Prakash et al, 2018; Benedicto et al, 2021). If untreated, most patients die at a mean age of 15 yrs because of myocardial infarction. Interestingly, CVD progression in HGPS closely resembles the geriatric-linked vascular pathology (Olive et al, 2010), therefore posing a valuable model for studying the underlying mechanisms of age-related CVD in the absence of classical risk factors such as hypercholesterolemia and high blood pressure usually seen in older patients (Bays et al, 2021). CVD progression has been linked to arterial stiffening, which occurs in both the physiological (Kohn et al, 2015) and premature HGPS aging (Olive et al, 2010).

In order to understand arterial dysfunction in HGPS- and age-linked CVD, one has to elucidate the disease mechanisms in the different arterial cell types and their individual contributions to

[1]Max Perutz Labs, Vienna Biocenter Campus (VBC), Vienna, Austria    [2]Medical University of Vienna, Max Perutz Labs, Vienna, Austria    [3]Vienna BioCenter PhD Program, A Doctoral School of the University of Vienna and the Medical University of Vienna, Vienna, Austria    [4]Center for Pathobiochemistry and Genetics, Medical University of Vienna, Vienna, Austria

Correspondence: selma.osmanagic-myers@meduniwien.ac.at; roland.foisner@meduniwien.ac.at

the pathologies. The arterial wall is composed of three distinct layers: the tunica intima, the innermost layer that contains endothelial cells (ECs) that are in direct contact with the blood flow; the tunica media, made of vascular smooth muscle cells (VSMCs); and the tunica adventitia, the outermost layer with perivascular adipose tissue cells and fibroblasts. Importantly, EC dysfunction is considered an early marker and driver of CVD and atherosclerosis and is characterized by reduced nitric oxide (NO) production, increased oxidative stress and inflammation, altered angiogenesis, and increased vascular permeability and leukocyte adhesion (Bonetti et al, 2003; Gimbrone & Garcia-Cardena, 2016; Allbritton-King & Garcia-Cardena, 2023). Aging and cellular senescence are known drivers of EC dysfunction (Herrera et al, 2010), and interestingly, EC senescence associated with pro-inflammatory and pro-fibrotic pathways has also been reported in different HGPS models (Matrone et al, 2019; Osmanagic-Myers et al, 2019; Atchison et al, 2020; Manakanatas et al, 2022; Xu et al, 2022; Vakili et al, 2025). Some of these phenotypes have been linked to activated or dysregulated TGFβ- (Hamczyk et al, 2024), YAP/TAZ- (Barettino et al, 2024), or MRTFA- (Osmanagic-Myers et al, 2019) signaling in HGPS ECs. However, a direct causal role of EC dysfunction for CVD pathologies has only been demonstrated in certain HGPS models (Osmanagic-Myers et al, 2019; Sun et al, 2020), but not in others (Del Campo et al, 2019, 2020; Benedicto et al, 2024, 2025a, 2025b; Hamczyk et al, 2024). Another known inducer of EC dysfunction is the dysregulated activation of the unfolded protein response (UPR) (Lenna et al, 2014; Cimellaro et al, 2016), which in turn can promote the development of senescence (Pluquet et al, 2015; Abbadie & Pluquet, 2020; Koloko Ngassie et al, 2024). However, it is unknown if and how progerin expression in ECs can elicit endoplasmic reticulum (ER) stress directly.

The main vascular pathology observed in HGPS patients and in most HGPS mouse models is a severely affected tunica media with progressive loss of VSMCs, most likely through apoptosis (Stehbens et al, 1999, 2001; Ackerman & Gilbert-Barness, 2002; Varga et al, 2006; Hamczyk et al, 2018). Interestingly, progerin-expressing VSMCs have been shown to display a dysregulated activation of the UPR, and VSMC loss results at least in part from increased ER stress and UPR activation (Hamczyk et al, 2019).

The UPR is activated by accumulation of unfolded or misfolded proteins in the ER lumen to restore ER homeostasis. It consists of three pathways, initiated by distinct ER membrane sensors—Inositol-requiring enzyme 1 α (IRE1α), protein kinase R-like endoplasmic reticulum kinase (PERK), and activating transcription factor 6 (ATF6)—which inhibit protein synthesis, eliminate misfolded/unfolded proteins, increase the folding capacity of the ER, and promote overall cell survival (Almanza et al, 2019; Hetz et al, 2020). IRE1α binds to the ER-resident chaperone BiP under steady-state conditions, whereas accumulation of misfolded proteins titrates BiP from the IRE1 lumenal domain (Bertolotti et al, 2000), leading to its oligomerization and auto-phosphorylation (Karagoz et al, 2017; Kettel et al, 2024). This in turn activates IRE1α RNase activity to produce *Xbp1s* from *Xbp1* mRNA by noncanonical splicing. The active transcription factor XBP1s then translocates into the nucleus and up-regulates the expression of chaperones, ER-associated protein degradation (ERAD) components, and genes related to autophagy and lipid synthesis (Walter & Ron, 2011).

Accumulation of unfolded proteins also activates the second ER sensor PERK, leading to its auto-phosphorylation and consequently phosphorylation of the eukaryotic translation initiator factor 2α (eIF2α), which exerts a dual function by attenuating general protein synthesis whereas at the same time promoting the translation of the transcription factor ATF4. ATF4 translocates to the nucleus and up-regulates the expression of *Gadd34* (*Ppp1r15a*), involved in cell recovery from protein synthesis shutoff. Other targets include chaperones and apoptosis-, autophagy-, oxidative response-, and redox regulation-related genes (Walter & Ron, 2011). Finally, activation of the third ER stress sensor ATF6 by unfolded proteins leads to its translocation to the Golgi. There, ATF6 is proteolytically cleaved and its cytosolic domain (ATF6 [N]) translocates to the nucleus, where it controls the expression of chaperones and ERAD components, as well as *Xbp1* (Walter & Ron, 2011).

Although both progerin-expressing fibroblasts and VSMCs have been reported to show a dysregulated activation of the UPR (Hamczyk et al, 2019; Vidak et al, 2023), not much is known about the proteostatic state of progerin-expressing ECs, and how this might account for endothelial senescence development, EC dysfunction, and CVD in the disease context. In this study, we therefore systematically investigate the activation of all three UPR pathways in progerin-expressing ECs from an EC-specific progeria mouse model (*Prog-Tg*) that recapitulates several cardiovascular pathologies observed in HGPS patients (Osmanagic-Myers et al, 2019). We consistently show that constitutive progerin expression in the endothelium does not promote a robust activation of the UPR at steady-state conditions in lung and heart ECs, whereas progerin-expressing ECs retain the ability to respond to ER stress when challenged. Moreover, we demonstrate that short-term acute progerin expression in ECs does initially activate the IRE1α pathway of the UPR, but that there is likely an adaptive mechanism that can down-regulate UPR activation back to basal levels upon constitutive progerin expression, as seen in HGPS arteries. Moreover, we found activation of the UPR pathways in arteries derived from an ubiquitous progerin-expressing mouse model, but not in arteries from the endothelium-specific model, demonstrating that the reported dysregulated up-regulation of the UPR in HGPS aorta (Hamczyk et al, 2019; Vidak et al, 2023) originates from non-endothelial arterial cell populations. Analysis of a published single-cell RNA-Seq (scRNA-Seq) dataset obtained from aortas of the ubiquitous progerin-expressing mouse model (Barettino et al, 2024) suggested activation of UPR signaling in aortic VSMCs, whereas fibroblasts and ECs showed no comparable changes within the resolution of this analysis. Overall, our data indicate that increased UPR activation is a cell-type-specific phenomenon in progerin-expressing arteries.

## Results

### Magnetic bead sorting allows for selective separation of ECs and non-ECs from the lungs of mice

Previous studies have shown that progerin-expressing VSMCs and fibroblasts display a dysregulated UPR activation (Hamczyk et al, 2019; Vidak et al, 2023). Although several studies have pointed to a

link between age-related EC dysfunction and ER stress (Battson et al, 2017; Ren et al, 2021), it remained unclear whether ECs in HGPS may also show UPR up-regulation.

To investigate proteostatic stress in progerin-expressing ECs, we isolated and purified ECs from lung tissue of 8–21 d-old *Prog-Tg* mice, which express progerin exclusively in the endothelium (Osmanagic-Myers et al, 2019). We used lung rather than cardiac or arterial tissue for the in vitro cell culture experiments, as ECs can be obtained from the lung in larger quantities and expanded in culture more efficiently compared with the other tissues. Furthermore, we previously showed that both the lung and heart ECs isolated from *Prog-Tg* mice have the same senescence phenotype associated with pro-inflammatory and pro-fibrotic signaling (Manakanatas et al, 2022; Fleischhacker et al, 2024). We used ICAM-2-coated magnetic beads to separate ECs from non-endothelial cells and confirmed successful sorting by flow cytometric analysis of the GFP signal, which served as a *proxy* for progerin expression since the mutant human Lamin A minigene (LA$^{G608G}$) in *Prog-Tg* mice contains an internal ribosomal entry site-driven GFP sequence (Fig 1A). As expected, only ECs isolated from *Prog-Tg* mice expressed GFP, with minimal contamination with GFP-negative cells (non-ECs) (Fig 1B). qRT-PCR-based analysis of the endothelial cell-specific marker *Cd31* (also known as *Pecam1*) and of progerin confirmed selective *Pecam1* expression in ECs (Fig 1C), with progerin found exclusively in *Prog-Tg* ECs (Fig 1D and E). These observations were also confirmed at the protein level by immunofluorescence microscopy (Fig 1F and G). Altogether, these results show that primary EC populations isolated from lung tissue are pure, with only minor contamination of non-ECs.

## Constitutive progerin expression in endothelial cells does not lead to a robust activation of the UPR

Loss of proteostasis activates a series of adaptive mechanisms, collectively known as the UPR, trying to restore proteostasis and cell survival (Hetz, 2012). The UPR is activated by three ER membrane-resident protein sensors, the most conserved one being IRE1 (Mori, 2009). Binding of accumulated misfolded proteins to the IRE1 lumenal domain (Bertolotti et al, 2000) leads to its oligomerization and auto-phosphorylation (Karagoz et al, 2017; Kettel et al, 2024), which activates its RNase activity to produce *Xbp1s* from *Xbp1* mRNA by non-canonical splicing (Fig 2A). Transcription factor XBP1s up-regulates expression of target genes, including ER-resident chaperones *Hsp90b1* and *Hspa5*, and *Edem1*, involved in the ERAD pathway (Walter & Ron, 2011) (Fig 2A). In primary lung ECs derived from *Prog-Tg* mice, we observed only a minor up-regulation of *Xbp1s* at the transcriptional level in *Prog-Tg* versus *WT* ECs, but downstream targets either remained unchanged (*Hspa5* and *Edem1*) or were even slightly down-regulated (*Hsp90b1*) (Fig 2B). Given these results, we hypothesized that the marginal increase in *Xbp1s* mRNA levels may not translate into increased levels of transcriptionally active XBP1s protein. Accordingly, XBP1s protein levels were unchanged in *Prog-Tg* versus *WT* ECs as measured by flow cytometry (Fig 2C), and immunofluorescence microscopy revealed only low, barely detectable XBP1s signals in the nucleus of both *WT* and *Prog-Tg* ECs (Fig 2D). Altogether, these data suggest that despite a minor up-regulation of

*Xbp1s* mRNA transcripts, this does not lead to a robust downstream signaling of the IRE1-mediated UPR pathway in lung ECs of *Prog-Tg* mice.

We next assessed the PERK pathway. PERK activation drives auto-phosphorylation and consequently phosphorylation of eIF2α, which attenuates general protein synthesis but at the same time promotes translation of transcription factor ATF4. ATF4 up-regulates expression of pro-apoptotic *Chop* (*Ddit3*) and of *Gadd34*, involved in cell recovery from protein synthesis shutoff (Walter & Ron, 2011) (Fig 3A). *Atf4* transcript levels (Fig 3B), and ATF4 protein levels, and ATF4 cellular localization, measured by flow cytometry (Fig 3C) and immunofluorescence microscopy (Fig 3D), were unchanged in *Prog-Tg* versus *WT* ECs. Accordingly, transcript levels of ATF4 target gene *Chop* were unchanged (Fig 3B). Only *Gadd34* was up-regulated in *Prog-Tg* versus *WT* ECs (Fig 3B), but *Gadd34* is also involved in the development of senescence (Yagi et al, 2003; Minami et al, 2007; Kalinin et al, 2023; Tomiyoshi et al, 2023). Overall, we concluded that the PERK pathway is not robustly up-regulated in *Prog-Tg* ECs.

Finally, we assessed the ATF6 branch of the UPR, in which activation of the ATF6 ER stress sensor leads to its translocation to the Golgi and subsequent proteolytic cleavage, producing transcription factor ATF6 (N) that controls expression of the ER-resident chaperone *Calr*, as well as *Xbp1*, *Hsp90b1*, *Hspa5*, and *Edem1*, also targeted by the IRE1 pathway (Walter & Ron, 2011) (Fig 3E). We neither observed changes in *Atf6* transcript levels in *Prog-Tg* versus *WT* ECs (Fig 3F), nor in the levels of the ATF6 target genes *Xbp1*, *Hsp90b1*, *Hspa5*, and *Edem1* (see Fig 2B), except for a minor up-regulation of *Calr* (Fig 3F). Thus, these data indicate that the ATF6 pathway is not activated in *Prog-Tg* ECs.

Overall, these results show that constitutive progerin expression in lung ECs does not lead to robust activation of any of the three UPR pathways, in stark contrast to VSMCs and fibroblasts (Hamczyk et al, 2019; Vidak et al, 2023).

## Progerin expression in endothelial cells does not impair the ability to respond to ER stress

Previous reports showed that progerin-expressing HGPS fibroblasts with an activated UPR are unable to further up-regulate UPR pathways upon applying additional ER stress (Vidak et al, 2023). We were therefore prompted to test if primary *Prog-Tg* ECs, which do not present robust up-regulation of UPR pathways, may respond to the artificial induction of ER stress. We challenged *WT* and *Prog-Tg* ECs with the pharmaceutical ER stress inducer Tunicamycin, which impairs N-glycosylation of proteins, leading to the accumulation of misfolded proteins and to ER stress. Initially, we performed a time course analysis by incubating *WT* ECs with 2.5 μg/ml Tunicamycin, known to induce ER stress in most cell types (Oslowski & Urano, 2011), for 0, 2, 4, 8, and 24 h, and tested transcript levels of *UPR* genes by qRT-PCR (Fig S1A). Effectors of each pathway (*Xbp1s*, *Atf4*, and *Atf6*) and the downstream target *Chop* showed progressive up-regulation during the incubation period, reaching a maximum level at around 8 h, followed by a decline after 24 h of treatment. The downstream targets *Hsp90b1* and *Calr* showed continuous up-regulation throughout the 24 h incubation. We found similar expression kinetics in *Prog-Tg* ECs (Fig S1B), indicating that they

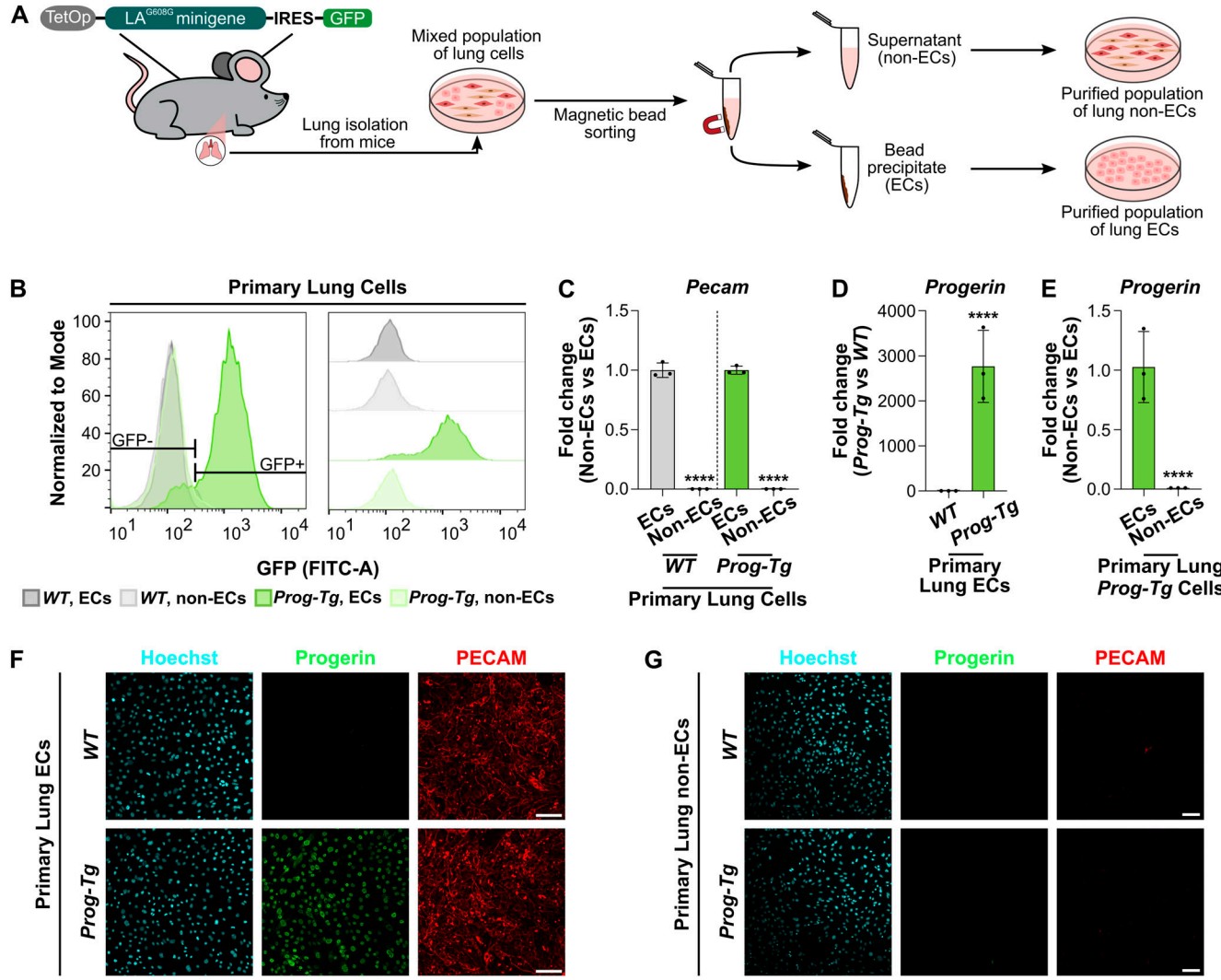

**Figure 1. Progerin expression is specific to endothelial cells of *Prog-Tg* mice.**
**(A)** Schematic representation of the workflow used to isolate ECs and non-ECs from lung tissue. *Prog-Tg* mice express a transgene containing a mutant human Lamin A minigene (LA^G608G) under the control of a tetracycline-responsive operator (TetOp), as well as a GFP-coding sequence expressed via an internal ribosomal entry site (Sagelius et al, 2008; Osmanagic-Myers et al, 2019). A mixed cell population was isolated from lungs of *WT* or *Prog-Tg* mice, which was further separated into ECs and non-ECs by sorting with ICAM-2-coated magnetic beads. **(B)** Flow cytometry analysis of GFP signal of primary lung ECs and non-ECs isolated from *WT* and *Prog-Tg* mice. Graphs were normalized to mode and smoothed. Gating for GFP⁻ and GFP⁺ cell populations are represented on the left, and offset graphs are shown on the right. 10,000 events were analyzed for each sample, and gated sequentially for live cells and single-cell populations (n = single live cells, $n_{WT, ECs}$ = 5,149, $n_{WT, non-ECs}$ = 6,774, $n_{Prog-Tg, ECs}$ = 6,549, $n_{Prog-Tg, non-ECs}$ = 6,533). **(C)** qRT-PCR analysis of *Pecam* expression in primary lung ECs and non-ECs isolated from *WT* and *Prog-Tg* mice, normalized to the reference gene *Hprt*, and to EC samples (n = 3 biological replicates, ****$P$ < 0.0001). **(D, E)** qRT-PCR analysis of progerin expression in primary lung ECs and non-ECs isolated from *WT* and *Prog-Tg* mice, normalized to the reference gene *Hprt*. **(D, E)** Data were further normalized to *WT* samples (D) or to EC samples (E) (n = 3 biological replicates, ****$P$ < 0.0001). **(F, G)** Representative immunofluorescence images of primary lung ECs (F) and non-ECs (G) isolated from *WT* and *Prog-Tg* mice, stained with anti-human LMNA (progerin) and anti-PECAM antibodies, and Hoechst. Scale bar = 100 µm. **(C, D, E)** Data information: In (C, D, E), all results are shown as mean ± SD. Significance was calculated by unpaired two-tailed *t* test.
Source data are available for this figure.

retain the ability to respond to external ER stressors. To test the extent of the ER stress response in *Prog-Tg* versus *WT* ECs, we analyzed the levels of the UPR-associated genes after Tunicamycin treatment at the 8 h timepoint, which corresponds to the peak of the UPR. All tested UPR genes were up-regulated at the transcriptional level upon Tunicamycin treatment in both *WT* (Fig 4A) and *Prog-Tg* (Fig 4B) ECs, and there was no difference in the extent of gene up-regulation in *Prog-Tg* versus *WT* ECs (Fig 4C). XBP1s and ATF4 were also similarly up-regulated on the protein level after

Tunicamycin treatment, as detected by flow cytometry (Fig 4D). Furthermore, Tunicamycin induced a similar increase in XBP1s protein levels in the nucleus in both *WT* and *Prog-Tg* ECs, as shown by immunofluorescence microscopy (Fig 4E) and quantification of nuclear XBP1s signal intensities (Fig 4F).

Altogether, these results demonstrate that *Prog-Tg* ECs are capable of activating UPR in response to ER stress to the same extent as *WT* cells. This suggests that there is no general impairment of the UPR pathways in lung ECs of *Prog-Tg* mice, contrary

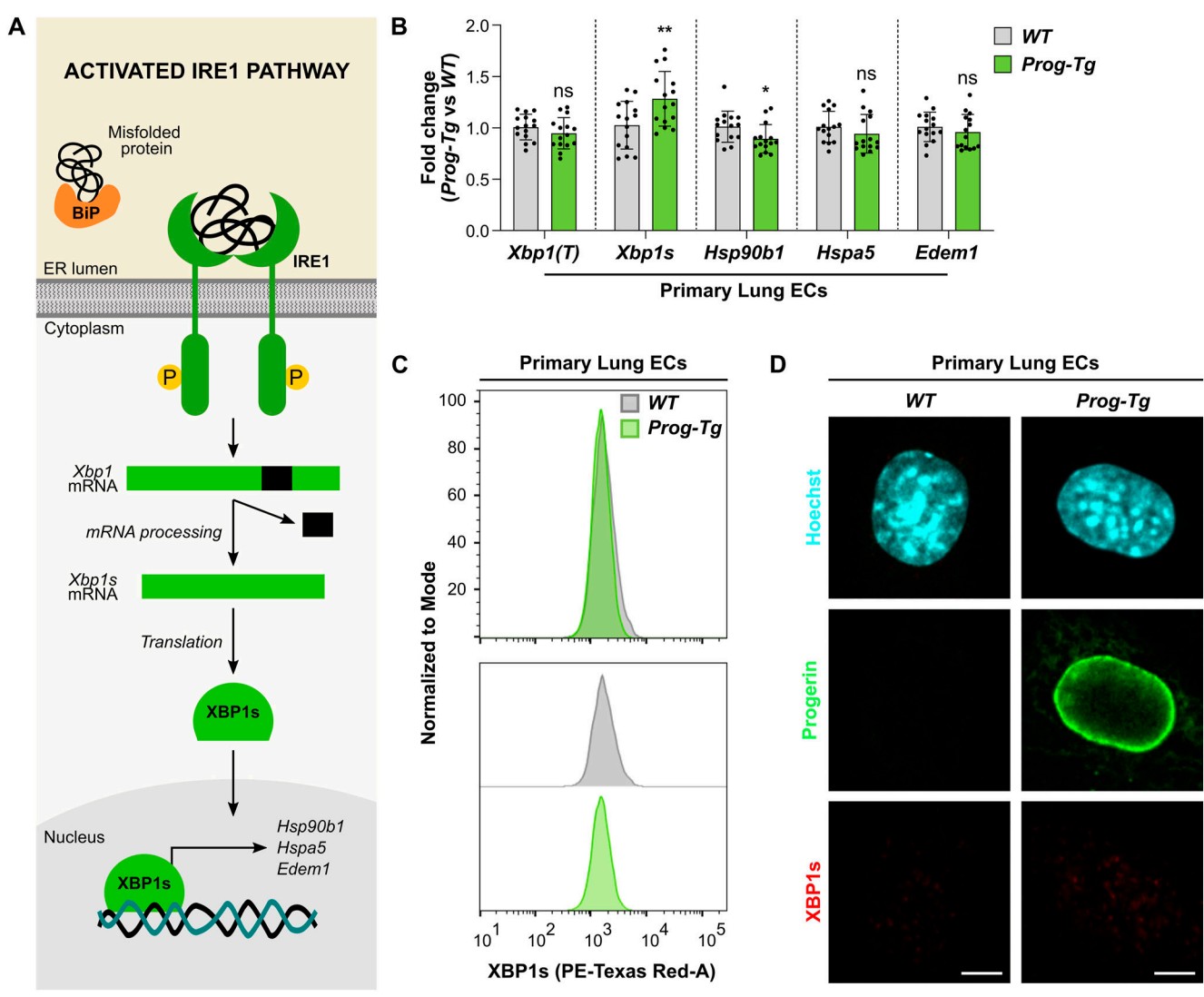

**Figure 2. The IRE1a pathway of the unfolded protein response (UPR) is not activated in endothelial cells of *Prog-Tg* mice.**
**(A)** Schematic representation of the activated IRE1α pathway of the UPR. Accumulated misfolded proteins in the ER lumen bind to IRE1α, the ER membrane sensor, causing dimerization and auto-phosphorylation. This activates its RNase activity, leading to unconventional splicing of *Xbp1* mRNA into *Xbp1s*. Transcription factor XBP1s translocates into the nucleus and up-regulates the expression of UPR target genes such as chaperones *Hsp90b1* and *Hspa5* (also known as BiP), and *Edem1*, involved in ER-associated protein degradation (ERAD). **(B)** qRT-PCR analysis of *Xbp1(T)* (unspliced and spliced isoforms), *Xbp1s*, *Hsp90b1*, *Hspa5*, and *Edem1* expression in primary lung ECs of *WT* and *Prog-Tg* mice. Expression levels were normalized to the reference gene *Hprt*, and to *WT* samples (n = 15 biological replicates) (ns $P_{Xbp1(T)}$ = 0.2223, **$P_{Xbp1s}$ = 0.0082, *$P_{Hsp90b1}$ = 0.0282, ns $P_{Hspa5}$ = 0.2278, ns $P_{Edem1}$ = 0.3515). **(C)** Flow cytometry analysis of primary lung ECs isolated from *WT* and *Prog-Tg* mice, stained with an anti-XBP1s antibody. Graphs were normalized to mode and smoothed, and offset graphs are shown at the bottom. Around 25,000 events were analyzed for each genotype, and gated sequentially for live cells, single-cell populations, and GFP$^-$/GFP$^+$ cells for *WT*/*Prog-Tg* samples (n = single live cells, n $_{WT}$ = 10,156, n $_{Prog-Tg}$ = 8,474). **(D)** Representative immunofluorescence images of primary lung ECs isolated from *WT* and *Prog-Tg* mice, stained with anti-human LMNA (progerin) and anti-XBP1s antibodies, and Hoechst. Scale bar = 5 *µ*m. **(B)** Data information: In (B), results are shown as mean ± SD and significance was calculated after batch correction by unpaired two-tailed *t* test.
Source data are available for this figure.

### Acute progerin expression in endothelial cells induces expression of XBP1s in the nucleus

A recent study using doxycycline-inducible expression of progerin in fibroblasts proposed that initial activation of UPR pathways results from the increasing accumulation and aggregation of

to what has been proposed in progerin-expressing VSMCs and fibroblasts (Hamczyk et al, 2019; Vidak et al, 2023).

progerin at the nuclear periphery, which may eventually lead to chronic ER stress and a dysregulated up-regulation of the UPR after constitutive progerin expression in HGPS fibroblasts (Vidak et al, 2023). Although *Prog-Tg* ECs also express progerin constitutively and are thus exposed to chronic stress, they do not show a dysregulated up-regulation of the UPR. Therefore, we hypothesized that *Prog-Tg* ECs may adapt to those conditions and attenuate UPR downstream signaling, as shown before for other cell types (Lin et al, 2007; Walter & Ron, 2011; Gomez & Rutkowski, 2016; Chen et al,

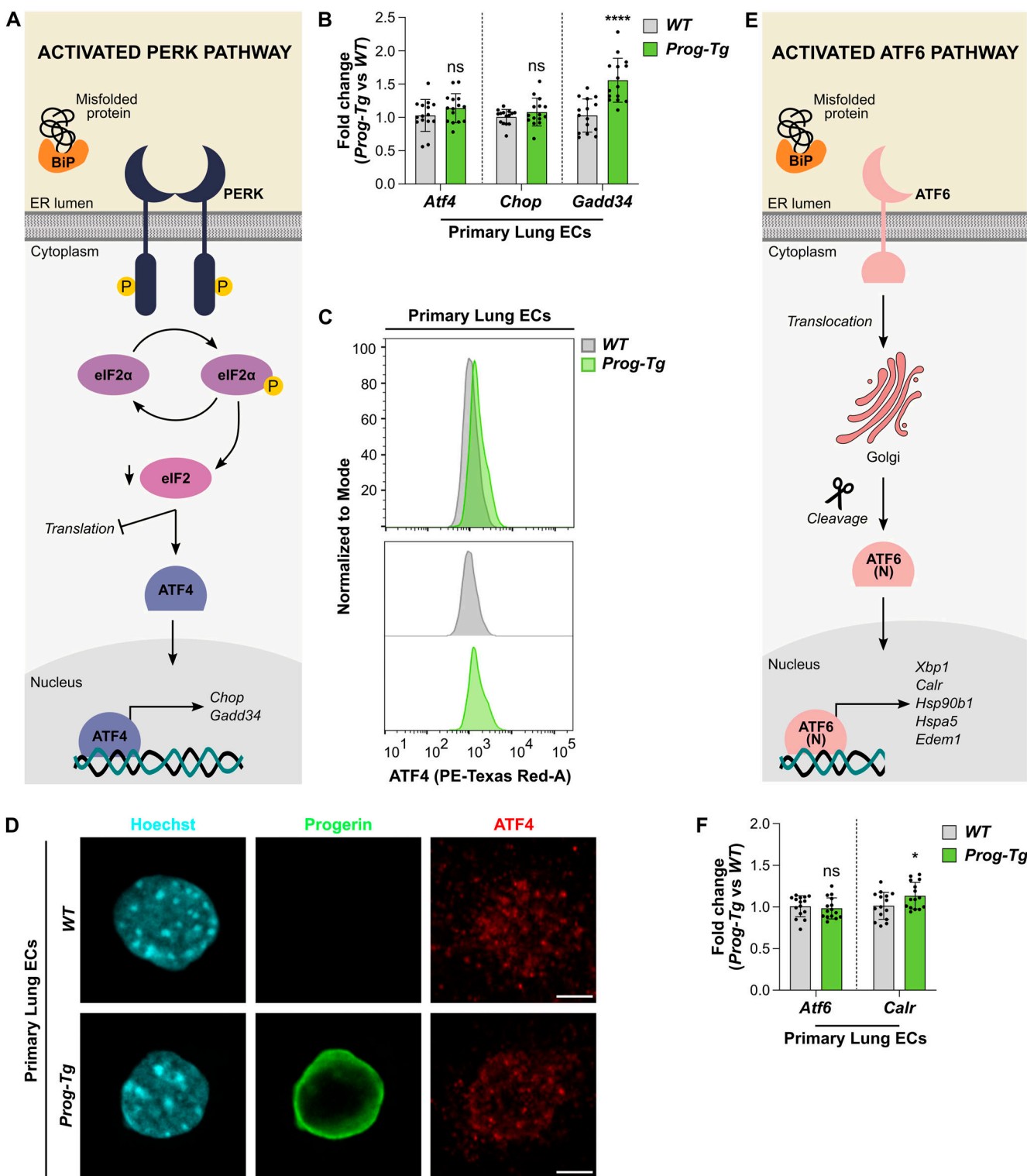

**Figure 3. The PERK and ATF6 pathways of the unfolded protein response (UPR) are not activated in *Prog-Tg* endothelial cells.**
**(A)** Scheme of the activated PERK branch of the UPR. When misfolded proteins accumulate, PERK dimerizes and auto-phosphorylates. PERK then phosphorylates the initiation factor eukaryotic translation initiator factor 2α (eIF2α), which exerts a dual function by attenuating general protein synthesis while at the same time promoting the translation of *Atf4*. ATF4 transcription factor translocates to the nucleus and targets the expression of genes such as pro-apoptotic *Chop*, and *Gadd34*, involved in cell recovery from protein synthesis shutoff. **(B)** qRT-PCR analysis of *Atf4*, *Chop* and *Gadd34* expression in primary lung ECs isolated from *WT* and *Prog-Tg* mice. Expression levels were normalized to the reference gene *Hprt*, and to *WT* samples (n = 15 biological replicates) (ns $P_{Atf4}$ = 0.1845, ns $P_{Chop}$ = 0.3283, ****$P_{Gadd34}$ < 0.0001). **(C)** Flow cytometry analysis of primary *WT* and *Prog-Tg* lung ECs, stained with an anti-ATF4 antibody. Graphs were normalized to mode and smoothed, and offset graphs are shown at the bottom. 10,000 events were analyzed for each genotype, and gated sequentially for live cells, single-cell populations, and GFP⁻/GFP⁺ cells for *WT*/*Prog-Tg* samples (n = single live cells, n $_{WT}$ = 6,365, n $_{Prog-Tg}$ = 2,311). **(D)** Representative immunofluorescence images of primary lung ECs isolated from *WT* and *Prog-Tg* mice,

2022). To test whether an acute expression of progerin in ECs may activate the UPR, we devised a strategy to analyze acute progerin expression in *Prog-Tg* ECs (Fig 5A). Doxycycline was added to the drinking water of *WT* and *Prog-Tg* mice, as well as to the cell culture medium of primary *WT* and *Prog-Tg* ECs, which inhibits expression of the doxycycline-dependent transactivator and the transactivator-driven progerin expression in *Prog-Tg* mice (Sun et al, 2005; Sagelius et al, 2008; Osmanagic-Myers et al, 2019). Analysis of the GFP signal intensity by flow cytometry (Fig 5B), as a *proxy* for progerin expression, confirmed that the progerin transgene was not expressed in *Prog-Tg* ECs cultured in the presence of doxycycline and within 2 d after doxycycline removal from the culture medium. Whereas we observed a subtle increase in GFP background signal upon doxycycline removal in all samples by flow cytometry, a positive GFP signal, clearly seen as a separate peak, was only detectable in *Prog-Tg* ECs starting 8 d after doxycycline removal, and the number of GFP-positive cells exponentially increased at later timepoints (Fig 5C). Similarly, progerin expression was clearly detectable in immunofluorescence microscopy 13 d after doxycycline removal (Fig 5D).

We next tested activation of the IRE1 pathway by assessing XBP1s nuclear localization after a 13 d doxycycline removal. *Prog-Tg* ECs with no progerin expression (+DOX) and *WT* ECs (+DOX and −DOX) did not express XBP1s in the nucleus, whereas *Prog-Tg* ECs with acute progerin expression (−DOX) had significantly increased XBP1s levels in the nucleus (Fig 5D and E). This indicates that the IRE1 pathway can be activated upon acute induction of progerin expression in ECs, whereas *Prog-Tg* ECs constitutively expressing progerin may indeed initiate an adaptive mechanism allowing them to attenuate UPR signaling. It has to be noted, however, that constitutive expression of progerin resembles the physiological conditions in HGPS rather than an acute/short-term progerin expression. Therefore, we concluded that ECs with constitutive progerin expression, as seen in *Prog-Tg* mice and in HGPS patients, can adapt to ER stress and attenuate UPR signaling.

### Absence of UPR signaling is a common characteristic of progerin-expressing ECs across tissues and ages

As endothelial cells are heterogeneous across tissues (Trimm & Red-Horse, 2023), we wondered whether, unlike in lung ECs, ER stress and UPR activation may be present in progerin-expressing ECs in organs that are physiologically relevant for CVD in HGPS, such as the heart and arteries. Moreover, CVD becomes more pronounced in old HGPS patients (Olsen et al, 2023) and mice (Kim et al, 2025). In order to address the tissue- and age-related aspects, we isolated ECs from the hearts of older mice (6 mo-old). Analyses of transcript levels of the EC-specific marker *Pecam* revealed enrichment for ECs in the EC versus non-EC fractions (Fig 6A, left

panel). We then tested UPR signaling in the EC population by qRT-PCR analyses. Similar to our results in lung ECs isolated from young mice, we did not observe up-regulation of genes related to the three UPR pathways in heart ECs derived from older *Prog-Tg* versus *WT* mice (Fig 6A, right panel).

To test UPR signaling in an in vivo context of whole tissues relevant for CVD, we analyzed UPR gene expression in lysates of heart tissue derived from 6 mo-old animals. Also, here no up-regulation of UPR-linked gene expression in *Prog-Tg* versus *WT* mice could be detected (Fig 6B). As several studies have described activation of UPR, particularly in the aorta of progerin-expressing mice (Hamczyk et al, 2019; Vidak et al, 2023), we tested aortic tissue isolated from 3.5 mo-old *Prog-Tg* and *WT* mice. However, no evident up-regulation of UPR signaling in the aorta of *Prog-Tg* versus *WT* mice was detected (Fig 6C). Overall, our results demonstrate that the absence of ER stress and UPR activation is a common feature of ECs constitutively expressing progerin, regardless of the tissue source and age.

### Activation of the UPR in aorta of HGPS mice is a cell-type-specific phenotype exclusively found in progerin-expressing VSMCs

Whereas previous studies claimed that the UPR is activated in progerin-expressing VSMCs and fibroblasts and in the aorta of HGPS mice (Hamczyk et al, 2019; Vidak et al, 2023), our results presented here clearly show no robust up-regulation of the three UPR pathways in progerin-expressing ECs across different tissues, including the lung, heart, and aorta. As previous studies have reported heterogeneous responses of different cells to ER stress (Reich et al, 2020; Anisimova & Karagoz, 2025; Gigan et al, 2025 *Preprint*), we reasoned that the observed activation of the UPR in HGPS aorta may be caused mostly by non-endothelial cells rather than ECs in the tissue.

To test this hypothesis, we used the transgenic *Lmna*$^{G609G/+}$ progeria mouse model, which has ubiquitous progerin expression (Osorio et al, 2011). Whereas in the aorta of *Prog-Tg* mice, progerin is expressed exclusively in ECs, in *Lmna*$^{G609G/+}$ mice, all aortic cell types, including ECs, VSMCs, fibroblasts, and adipose cells, express progerin (Fig 7A). Therefore, the aorta poses a great model to study the proteostatic state of non-ECs in an in vivo context.

We tested the expression of UPR genes in whole aorta lysates of *Lmna*$^{G609G/+}$ mice by qRT-PCR and found that most genes were either significantly up-regulated or showed a clear trend for up-regulation compared with *Lmna*$^{+/+}$ control samples (Fig 7B). As expected, these genes were not up-regulated in aortic lysates from *Prog-Tg* mice lacking progerin expression in non-endothelial cells (see Fig 6C). These results strongly suggest that the up-regulation of the UPR pathways in *Lmna*$^{G609G/+}$ versus *Prog-Tg* aortas is caused by progerin-expressing non-endothelial cells present only in *Lmna*$^{G609G/+}$

---

stained with anti-human LMNA (progerin) and anti-ATF4 antibodies, and Hoechst. Scale bar = 5 μm. **(E)** Schematic representation of the activated ATF6 branch of the UPR. Upon accumulation of misfolded proteins in the ER lumen, ATF6 is transported to the Golgi, where it is proteolytically cleaved. ATF6 cytosolic domain (ATF6 [N]) translocates to the nucleus and controls the expression of genes such as the chaperone *Calr*, as well as *Xbp1*, *Hsp90b1*, *Hspa5*, and *Edem1*, all common to the IRE1 pathway of the UPR. **(F)** qRT-PCR analysis of *Atf6* and *Calr* expression in primary lung ECs from *WT* and *Prog-Tg* mice. Expression levels were normalized to the reference gene *Hprt*, and to *WT* samples (n = 15 biological replicates) (ns P$_{Atf6}$ = 0.5989, *P$_{Calr}$ = 0.0454). **(B, F)** Data information: In (B, F), results are shown as mean ± SD. Significance was calculated after batch correction by unpaired two-tailed *t* test.
Source data are available for this figure.

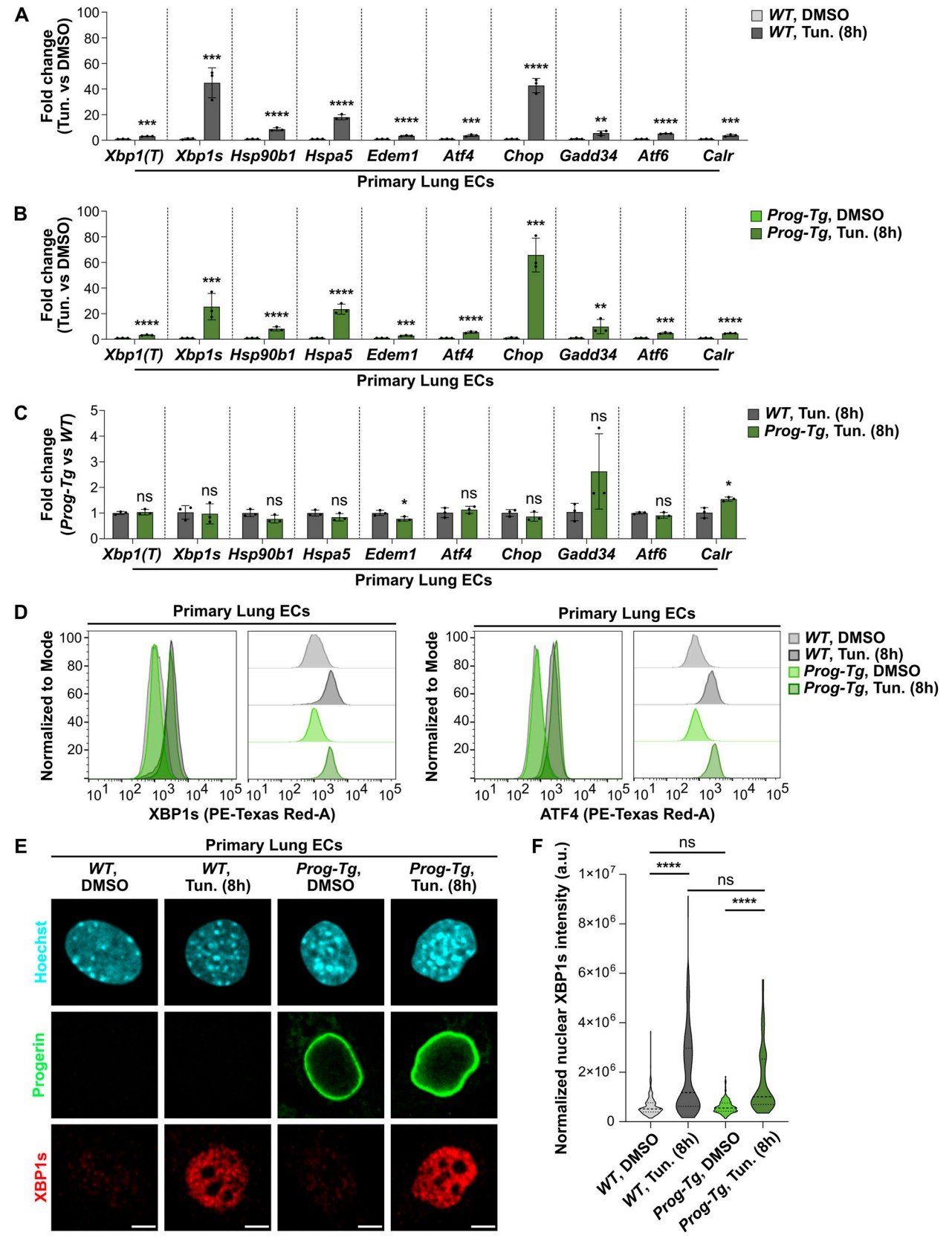

**Figure 4. Progerin-expressing endothelial cells are able to respond to increased ER stress by up-regulating all branches of the unfolded protein response.**
**(A, B)** qRT-PCR analysis of *Xbp1(T)*, *Xbp1s*, *Hsp90b1*, *Hspa5*, *Edem1*, *Atf4*, *Chop*, *Gadd34*, *Atf6*, and *Calr* expression in primary lung ECs isolated from *WT* (A) and *Prog-Tg* (B) mice. Cells were treated with Tunicamycin (2.5 μg/ml, 8 h) and the respective DMSO control. Expression levels were normalized to the reference gene *Hprt* and to

samples, whereas progerin-expressing ECs present in both mouse models do not contribute to elevated UPR in aortic tissue.

Accordingly, ECs isolated from lung of $Lmna^{G609G/+}$ mice do not have elevated levels of UPR-related genes ($Xbp1s$, $Edem1$, $Gadd34$, $Atf6$, $Calr$) compared with ECs from $Lmna^{+/+}$ mice, and some genes ($Xbp1(T)$, $Hsp90b1$, $Hspa5$, $Atf4$, $Chop$) were even down-regulated in $Lmna^{G609G/+}$ versus $Lmna^{+/+}$ ECs (Fig 7C). Thus, progerin-expressing lung ECs derived from two different HGPS mouse models ($Prog$-$Tg$ and $Lmna^{G609G/+}$ mice) do not show robust activation of the UPR. Immunoblot analyses of whole lysates of these ECs using an anti-mouse and anti-human lamin A antibody detected the shorter lamin A disease variant, progerin in both the $Lmna^{G609G/+}$ and $Prog$-$Tg$ cells, similar to a HGPS patient fibroblast sample, whereas it was absent in $WT$ ECs (Fig 7D). Protein levels of ectopically expressed human progerin in $Prog$-$Tg$ ECs seem to be higher than the endogenous mouse progerin in $Lmna^{G609G/+}$ ECs (Fig 7D). Despite the lower progerin expression level in $Lmna^{G609G/+}$ versus $Prog$-$Tg$ samples, aortic samples from $Lmna^{G609G/+}$ but not from $Prog$-$Tg$ mice showed elevated UPR signaling (see Figs 6C and 7B), reinforcing the hypothesis that loss of proteostasis in aortic tissue originates from arterial non-endothelial cells.

To further confirm a lack of UPR activation in aortic ECs of ubiquitous progerin-expressing mice, and to identify cell types in the aorta with a dysregulated ER stress pathway, we analyzed a publicly available scRNA-Seq dataset from whole aortas of older $Lmna^{G609G/G609G}$ and $Lmna^{+/+}$ mice (Barettino et al, 2024). We first compared the transcript levels of the previously analyzed set of genes representing all three UPR pathways in the combined VSMC-, fibroblast-, and EC-populations described in the previous study. We found up-regulation of the PERK pathway only in the combined aortic VSMC population in $Lmna^{G609G/G609G}$ versus $Lmna^{+/+}$ samples. In contrast, aortic fibroblast- and EC-populations showed no up-regulation of the UPR pathways (Fig 8A). This suggests that loss of proteostasis in progeroid aorta is likely rooted in progerin-expressing aortic VSMCs and presumably fibroblasts. The previous study also reported a subpopulation of dysfunctional VSMCs in the aortas of $Lmna^{G609G/G609G}$ mice, whose scRNA-Seq dataset was enriched in gene ontology (GO) terms related to ER stress, UPR, and apoptosis (Barettino et al, 2024). Examination of these GO term-associated differentially expressed genes in the merged

VSMC-, fibroblast-, and EC populations indicated that UPR-related genes were mainly up-regulated in aortic VSMCs, with smaller increases detected in fibroblasts, in $Lmna^{G609G/G609G}$ versus $Lmna^{+/+}$ samples (Fig 8B). Most of these genes were only slightly increased, unchanged, or down-regulated in combined aortic EC populations in $Lmna^{G609G/G609G}$ versus $Lmna^{+/+}$ samples, with the exception of $Thbs1$, which was highly increased in progerin-expressing ECs versus $WT$ cells. This may be explained by the important roles of the $Thbs1$ gene in normal EC function and angiogenesis (Liu et al, 2024). Lastly, given that VSMC loss is a hallmark of CVD in HGPS (Hamczyk & Andrés, 2019), we analyzed the expression of ER stress-dependent apoptotic gene sets, which were also reported in the subpopulation of dysfunctional VSMCs in the aorta of $Lmna^{G609G/G609G}$ mice. We found these genes prominently up-regulated also in combined VSMC- and fibroblast-populations in $Lmna^{G609G/G609G}$ versus $Lmna^{+/+}$ samples, whereas only a few of them were slightly up in EC populations (Fig 8C).

Altogether, these data confirm our observations on a lack of a robust UPR in progerin-expressing ECs and indicate that up-regulation of the UPR reported in the aorta of HGPS mice and patients are mostly caused by progerin-expressing non-endothelial cells, particularly VSMCs, rather than by ECs. Accordingly, we conclude that UPR in ECs does not contribute to EC dysfunction and CVD in HGPS.

## Discussion

Endothelial dysfunction represents an early event in the development of CVD (Allbritton-King & Garcia-Cardena, 2023) and is also a feature of the normal aging process (Herrera et al, 2010). In the premature aging disease HGPS, which mimics several features of normative aging, ECs have been shown to develop inflammation, fibrosis, and senescence (Matrone et al, 2019; Osmanagic-Myers et al, 2019; Atchison et al, 2020; Manakanatas et al, 2022; Xu et al, 2022; Vakili et al, 2025), all of which are associated with a dysfunctional endothelium. Moreover, in an HGPS mouse model with EC-specific progerin expression, several of the CVD phenotypes observed in patients were recapitulated, such as cardiac

DMSO-treated samples (n = 3 biological replicates) (***$P_{WT, Xbp1(T)}$ = 0.0001, ***$P_{WT, Xbp1s}$ = 0.0006, ****$P_{WT, Hsp90b1}$ < 0.0001, ****$P_{WT, Hspa5}$ < 0.0001, ****$P_{WT, Edem1}$ < 0.0001, ***$P_{WT, Atf4}$ = 0.0005, ****$P_{WT, Chop}$ < 0.0001, **$P_{WT, Gadd34}$ = 0.0018, ****$P_{WT, Atf6}$ < 0.0001, ***$P_{WT, Calr}$ = 0.0008, ****$P_{Prog-Tg, Xbp1(T)}$ < 0.0001, ***$P_{Prog-Tg, Xbp1s}$ = 0.0001, ****$P_{Prog-Tg, Hsp90b1}$ < 0.0001, ****$P_{Prog-Tg, Hspa5}$ < 0.0001, ***$P_{Prog-Tg, Edem1}$ = 0.0001, ****$P_{Prog-Tg, Atf4}$ < 0.0001, ***$P_{Prog-Tg, Chop}$ = 0.0002, **$P_{Prog-Tg, Gadd34}$ = 0.0025, ***$P_{Prog-Tg, Atf6}$ = 0.0003, ****$P_{Prog-Tg, Calr}$ < 0.0001). **(C)** qRT-PCR analysis of $Xbp1(T)$, $Xbp1s$, $Hsp90b1$, $Hspa5$, $Edem1$, $Atf4$, $Chop$, $Gadd34$, $Atf6$, and $Calr$ expression in Tunicamycin-treated (2.5 µg/ml, 8 h) ECs isolated from $WT$ and $Prog$-$Tg$ mice. Expression levels were normalized to the reference gene $Hprt$ and to $WT$ samples (n = 3 biological replicates) (ns $P_{Xbp1(T)}$ = 0.6617, ns $P_{Xbp1s}$ = 0.7727, ns $P_{Hsp90b1}$ = 0.1016, ns $P_{Hspa5}$ = 0.1759, *$P_{Edem1}$ = 0.0338, ns $P_{Atf4}$ = 0.4201, ns $P_{Chop}$ = 0.2921, ns $P_{Gadd34}$ = 0.0722, ns $P_{Atf6}$ = 0.2389, *$P_{Calr}$ = 0.0236). **(D)** Flow cytometry analysis of primary lung ECs from $WT$ and $Prog$-$Tg$ mice, stained with anti-XBP1s **(left)** and anti-ATF4 **(right)** antibodies. Cells were treated with Tunicamycin (2.5 µg/ml, 8 h) and the respective DMSO control. Graphs were normalized to mode and smoothed, and offset graphs are also shown. Around 15,000–40,000 events were analyzed for each genotype, and gated sequentially for live cells, single-cell populations, and GFP⁻/GFP⁺ cells for $WT$/$Prog$-$Tg$ samples (n = single live cells, n $_{WT, DMSO, XBP1s}$ = 20,772, n $_{WT, Tun. (8h), XBP1s}$ = 21,234, n $_{Prog-Tg, DMSO, XBP1s}$ = 9,946, n $_{Prog-Tg, Tun. (8h), XBP1s}$ = 10,502, n $_{WT, DMSO, ATF4}$ = 9,628, n $_{WT, Tun. (8h), ATF4}$ = 9,710, n $_{Prog-Tg, DMSO, ATF4}$ = 8,314, n $_{Prog-Tg, Tun. (8h), ATF4}$ = 8,602). **(E)** Representative immunofluorescence images of $WT$ and $Prog$-$Tg$ primary lung ECs, treated with Tunicamycin (2.5 µg/ml, 8 h) and the respective DMSO control. Cells were stained with anti-human LMNA (progerin) and anti-XBP1s antibodies, and Hoechst. Scale bar = 5 µm. **(F)** Quantification of nuclear XBP1s intensity, based on immunofluorescence images of $WT$ and $Prog$-$Tg$ ECs treated with Tunicamycin (2.5 µg/ml, 8 h) and the respective DMSO control. The macro described in the Materials and Methods section was used for the quantification (n = cells, n $_{WT, DMSO}$ = 186, n $_{WT, Tun. (8h)}$ = 171, n $_{Prog-Tg, DMSO}$ = 184, n $_{Prog-Tg, Tun. (8h)}$ = 137) (ns $P_{WT, DMSO \text{ versus } Prog-Tg, DMSO}$ > 0.9999, ns $P_{WT, Tun. (8 h) \text{ versus } Prog-Tg, Tun. (8 h)}$ > 0.9999, ****$P_{WT, DMSO \text{ versus } WT, Tun. (8 h)}$ < 0.0001, ****$P_{Prog-Tg, DMSO \text{ versus } Prog-Tg, Tun. (8 h)}$ < 0.0001). **(A, B, C, F)** Data information: in (A, B, C), results are shown as mean ± SD and significance was calculated by unpaired two-tailed $t$ test. In (F), significance was calculated by repeated-measures Kruskal-Wallis test.
Source data are available for this figure.

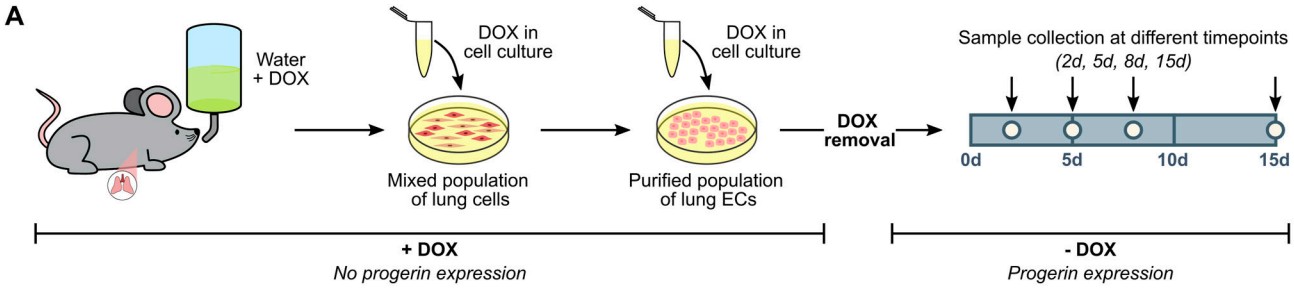

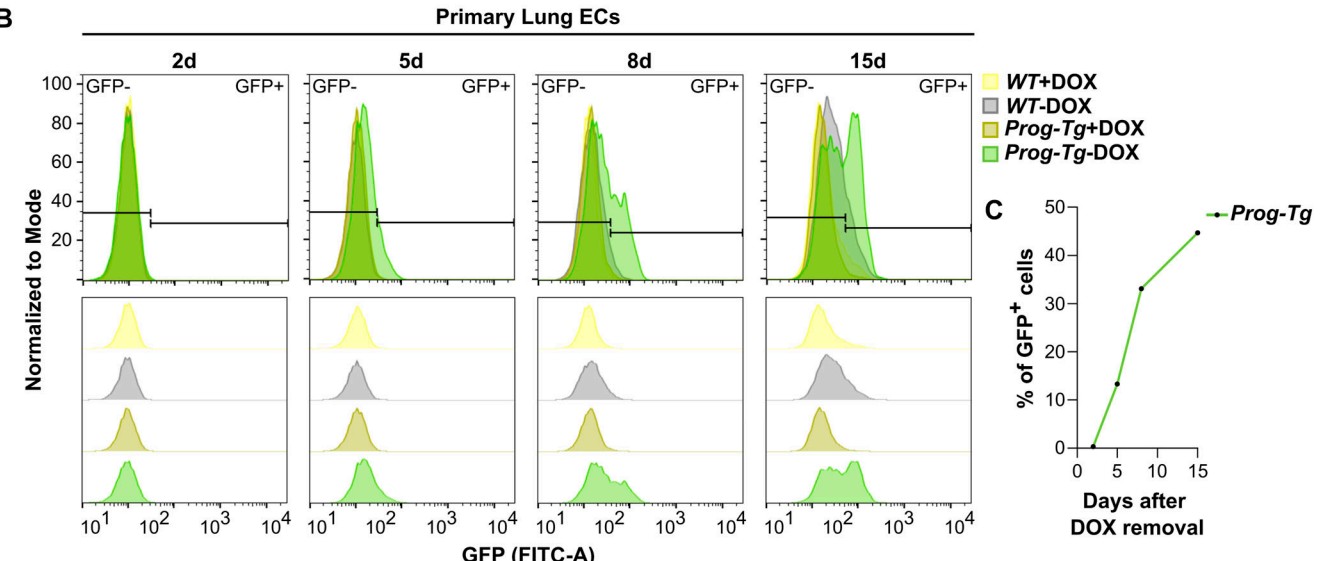

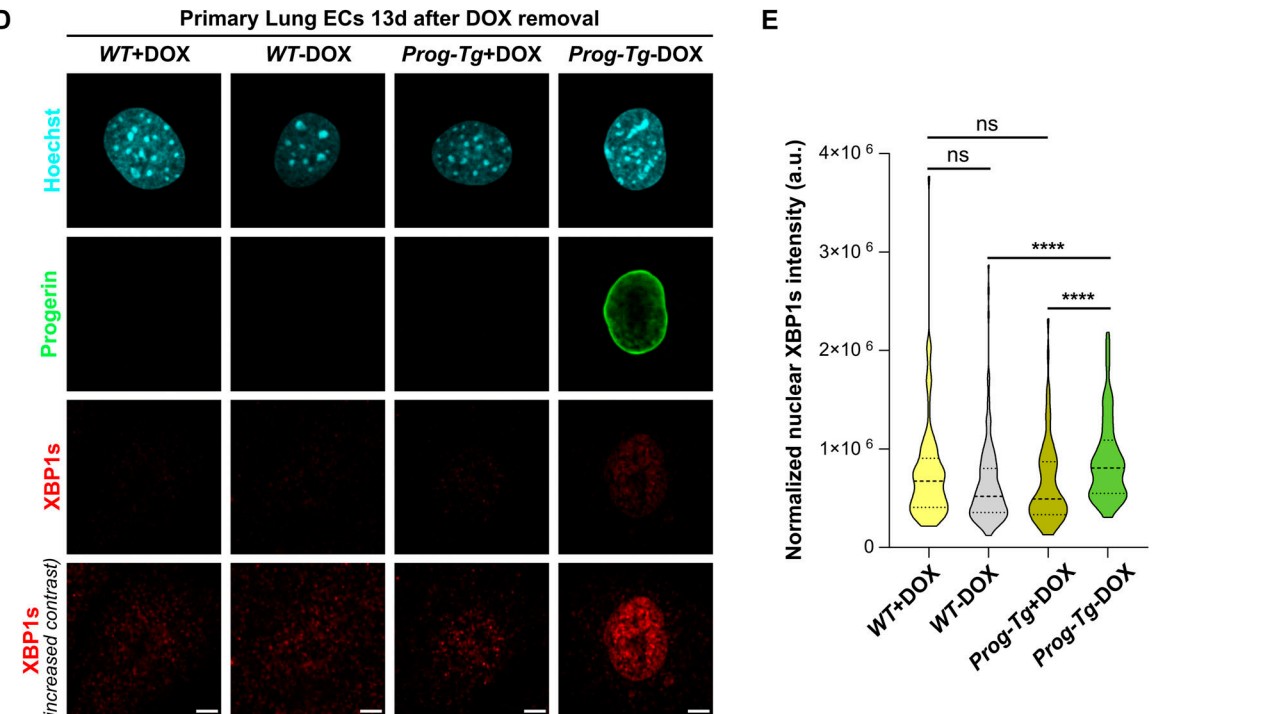

**Figure 5. Acute progerin expression activates the IRE1 signaling pathway in *Prog-Tg* endothelial cells.**

**(A)** Schematic representation of the workflow used to control progerin expression. Doxycycline hyclate (10 mg/ml) was added to the drinking water of mice, to prevent progerin expression from the beginning of development. Isolated primary cells were kept in doxycycline until sorting. Afterward, doxycycline was removed from the cell culture medium (day 0), and samples were collected at different timepoints after doxycycline removal (day 2, day 5, day 8, and day 15), to identify the moment progerin

hypertrophy, diastolic dysfunction, and interstitial fibrosis (Osmanagic-Myers et al, 2019). However, the molecular mechanisms by which progerin expression drives EC dysfunction and CVD development are still elusive. Given that loss of proteostasis is present in the aging endothelium (Gargalovic et al, 2006; Kassan et al, 2012; Battson et al, 2017) and is intimately related to CVD development (Ren et al, 2021), we set out to assess the proteostatic state of progerin-expressing ECs. In this study, we found no robust loss of proteostasis and activation of an ER stress response in primary ECs with progerin expression, isolated from differently aged mice and derived from different tissues (lung and heart). Also, in aortic tissue from *Prog-Tg* mice, which express progerin exclusively in the endothelium, we did not observe robust up-regulation of UPR pathways. In contrast, we detected an activated UPR in aorta derived from a *Lmna*$^{G609G/+}$ mouse model, in which all aortic cell types, including ECs, VSMCs, fibroblasts, and adipose tissue, express progerin. Analysis of a published scRNA-Seq dataset from aortic tissue (Barettino et al, 2024) suggested that UPR activation in the aorta of *Lmna*$^{G609G/G609G}$ mice is predominantly associated with VSMCs. Thus, the up-regulation of ER stress response pathways in the aortic tissue in HGPS seems to be cell-type-specific and exclusive to progerin-expressing non-endothelial cells, supporting the previous findings in fibroblasts (Vidak et al, 2023) and VSMCs (Hamczyk et al, 2019) from different HGPS models.

Impaired protein homeostasis has long been considered a hallmark of aging (Lopez-Otin et al, 2013, 2023) and is associated with enhanced production and accumulation of misfolded proteins and reduced waste removal capacity, which impairs the cells' ability to respond to ER stress. Interestingly, loss of proteostasis has also been reported in several in vitro and in vivo models of HGPS. Progerin expression in fibroblasts and VSMCs was associated with up-regulation of the UPR, accompanied by activation of all three ER membrane sensors, and increased expression of ER chaperones, ERAD-related genes, and apoptotic factors (Hamczyk et al, 2019; Vidak et al, 2023). Progerin-expressing fibroblasts were also unable to further up-regulate the expression of UPR-related genes when challenged with an ER stressor, as a result of the already dysregulated activation of these pathways (Vidak et al, 2023). The authors then proposed that progerin in the cell nucleus activates the ER stress response by clustering SUN2 at the inner nuclear membrane, consequently sequestering BiP to the nuclear periphery, and initiating the downstream signaling cascades associated with the UPR (Vidak et al, 2023). Although we have previously reported an accumulation of SUN2 at the nuclear periphery in *Prog-Tg* ECs (Osmanagic-Myers et al, 2019), we surprisingly found no consistent up-regulation of UPR-related markers. Moreover, *Prog-Tg* ECs had no impaired response to the ER stressor Tunicamycin, showing a similar transcriptional up-regulation of all tested UPR genes compared with *WT* ECs.

Our data, therefore, show that there is no robust activation of the UPR in *Prog-Tg* ECs constitutively expressing progerin, regardless of tissue origin or animal age. Given these surprising results, we hypothesized that the cell-type-specific differences in the response to proteostatic stress could explain the conflicting data. In fact, several studies have reported differential ER stress responses between different cell types and tissues (van Ziel & Scheper, 2020). In aged *C. elegans*, UPR-related genes were down-regulated in several non-neuronal cells, but specific subsets of neuronal cells had increased expression of chaperones (Roux et al, 2023). In a different study, XBP1s up-regulation under ER stress conditions led to the transcription of distinct target genes in skeletal muscle versus secretory cells (Acosta-Alvear et al, 2007). In aged mice, even different types of skeletal muscle cells showed differential activation of the UPR pathways (Chalil et al, 2015). Moreover, in an *ApoE*$^{-/-}$ mouse model, aortic tissue had increased UPR signaling with aging, whereas lung tissue remained unaffected (Zhou et al, 2021). In line with these previous findings, we found a robust activation of the UPR in aortas from older *Lmna*$^{G609G/+}$ mice, with ubiquitous progerin expression in all cell types, but not in aortas and hearts from older *Prog-Tg* mice, with EC-specific progerin expression. scRNA-Seq analysis of aortic VSMC-, fibroblast-, and EC-populations of *Lmna*$^{G609G/G609G}$ mice confirmed the lack of ER stress response in ECs, and instead revealed that the UPR up-regulation observed in aortic tissue originates mainly from non-endothelial cells with a strong ER stress-dependent pro-apoptotic signature. Altogether, our results show that the loss of proteostasis phenotype is not found in progerin-expressing ECs, regardless of its origin from EC-specific or ubiquitous progerin-expressing mice.

The cause for the observed differences in the response of ECs versus non-endothelial cells to ER stress is still unclear. The results obtained in this study seem to point to a model in which acute progerin expression may initially activate the UPR in both non-endothelial cells and ECs, as demonstrated in our experiments in ECs upon acute progerin expression, with different downstream outcomes in both cell types. Given that ECs are more resistant to stress and apoptosis compared with VSMCs (Shaw et al, 2021;

expression begins. **(B)** Flow cytometry analysis of GFP signal of *WT* and *Prog-Tg* primary lung ECs, kept in the presence or absence of doxycycline, at different timepoints after doxycycline removal from cell culture (2d, 5d, 8d, and 15d). Graphs were normalized to mode and smoothed. Gating for GFP⁻ and GFP⁺ cell populations are represented on the top graphs, and offset graphs are shown on the bottom. 10,000 events were analyzed for each sample, and gated sequentially for live cells and single-cell populations (n = single live cells, n $_{WT+DOX, 2d}$ = 5,818, n $_{WT+DOX, 5d}$ = 6,242, n $_{WT+DOX, 8d}$ = 5,870, n $_{WT+DOX, 15d}$ = 5,546, n $_{WT-DOX, 2d}$ = 5,637, n $_{WT-DOX, 5d}$ = 5,812, n $_{WT-DOX, 8d}$ = 4,376, n $_{WT-DOX, 15d}$ = 2,635, n $_{Prog-Tg+DOX, 2d}$ = 6,244, n $_{Prog-Tg+DOX, 5d}$ = 7,036, n $_{Prog-Tg+DOX, 8d}$ = 6,555, n $_{Prog-Tg+DOX, 15d}$ = 5,572, n $_{Prog-Tg-DOX, 2d}$ = 5,588, n $_{Prog-Tg-DOX, 5d}$ = 5,387, n $_{Prog-Tg-DOX, 8d}$ = 3,937, n $_{Prog-Tg-DOX, 15d}$ = 3,082). **(C)** Percentage of GFP-positive cells in *Prog-Tg* ECs after doxycycline removal (2d, 5d, 8d, and 15d), based on flow cytometry data shown in (B) (2d = 0.3%, 5d = 13.3%, 8d = 33.1%, 15d = 44.7%). **(D)** Representative immunofluorescence images of *WT* and *Prog-Tg* primary lung ECs, kept in the presence or absence of doxycycline. Cells were stained with anti-human LMNA (progerin) and anti-XBP1s antibodies, and Hoechst. The bottom panel shows the XBP1s staining with increased intensity for better visualization. Scale bar = 5 μm. **(E)** Quantification of nuclear XBP1s intensity, based on immunofluorescence images of *WT* and *Prog-Tg* primary lung ECs kept in the presence or absence of doxycycline. The macro described in the Materials and Methods section was used for the quantification (n = cells, n $_{WT+DOX}$ = 95, n $_{WT-DOX}$ = 146, n $_{Prog-Tg+DOX}$ = 77, n $_{Prog-Tg-DOX}$ = 90) (ns P $_{WT+DOX \ versus \ Prog-Tg+DOX}$ = 0.3165, ****P $_{WT-DOX \ versus \ Prog-Tg-DOX}$ < 0.0001, ns P $_{WT+DOX \ versus \ WT-DOX}$ = 0.1358, ****P $_{Prog-Tg+DOX \ versus \ Prog-Tg-DOX}$ < 0.0001). **(E)** Data information: in (E), significance was calculated by repeated-measures Kruskal-Wallis test. Source data are available for this figure.

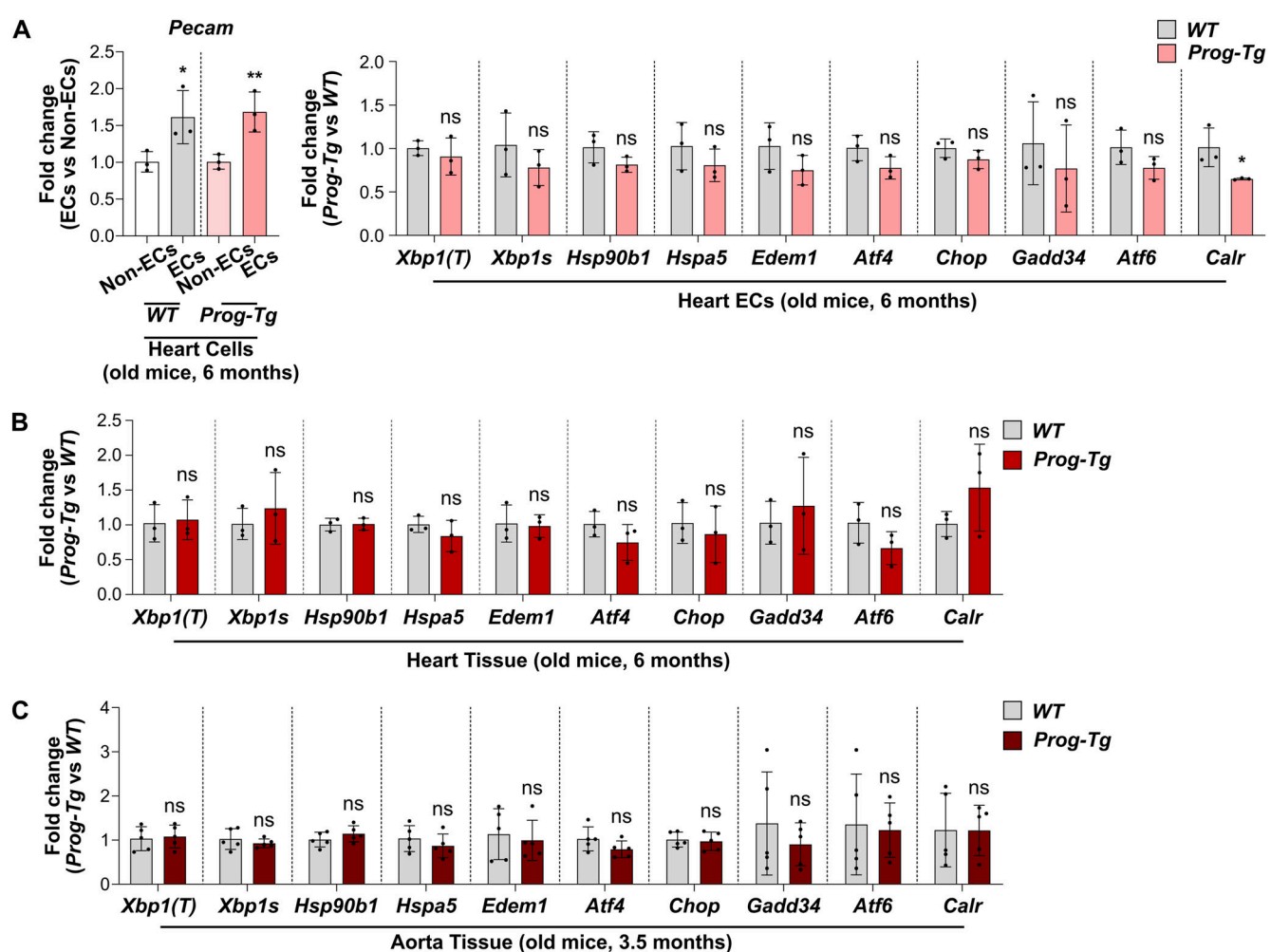

**Figure 6. The unfolded protein response pathways are not activated in heart ECs and heart and aorta tissue from older *Prog-Tg* mice.**
**(A)** qRT-PCR analysis of the EC-specific marker *Pecam* in non-ECs and ECs **(left)**, and the unfolded protein response-related genes *Xbp1(T), Xbp1s, Hsp90b1, Hspa5,* *Edem1, Atf4, Chop, Gadd34, Atf6,* and *Calr* in ECs **(right)**, sorted from heart tissue of older *WT* and *Prog-Tg* mice. Expression levels were normalized to the reference gene *Hprt*, and to non-EC **(left)** or *WT* **(right)** samples (n = 3 biological replicates) (*$P_{WT, Pecam}$ = 0.0338, **$P_{Prog-Tg, Pecam}$ = 0.0092, ns $P_{Xbp1(T)}$ = 0.4720, ns $P_{Xbp1s}$ = 0.3578, ns $P_{Hsp90b1}$ = 0.1616, ns $P_{Hspa5}$ = 0.3186, ns $P_{Edem1}$ = 0.2243, ns $P_{Atf4}$ = 0.1120, ns $P_{Chop}$ = 0.2015, ns $P_{Gadd34}$ = 0.4230, ns $P_{Atf6}$ = 0.1482, *$P_{Calr}$ = 0.0229). **(B)** qRT-PCR analysis of *Xbp1(T), Xbp1s, Hsp90b1, Hspa5, Edem1, Atf4, Chop, Gadd34, Atf6,* and *Calr* expression in whole heart tissue lysates from *WT* and *Prog-Tg* mice. Expression levels were normalized to the reference gene *Hprt*, and to *WT* samples (n = 3 mice) (ns $P_{Xbp1(T)}$ = 0.8178, ns $P_{Xbp1s}$ = 0.5999, ns $P_{Hsp90b1}$ = 0.8841, ns $P_{Hspa5}$ = 0.3130, ns $P_{Edem1}$ = 0.8830, ns $P_{Atf4}$ = 0.2451, ns $P_{Chop}$ = 0.5474, ns $P_{Gadd34}$ = 0.7287, ns $P_{Atf6}$ = 0.1896, ns $P_{Calr}$ = 0.2869). **(C)** qRT-PCR analysis of *Xbp1(T), Xbp1s, Hsp90b1, Hspa5, Edem1, Atf4, Chop,* *Gadd34, Atf6,* and *Calr* expression in whole aorta tissue lysates from *WT* and *Prog-Tg* mice. Expression levels were normalized to the reference gene *Hprt*, and to *WT* samples (n = 5 mice) (ns $P_{Xbp1(T)}$ = 0.7713, ns $P_{Xbp1s}$ = 0.5114, ns $P_{Hsp90b1}$ = 0.2589, ns $P_{Hspa5}$ = 0.3680, ns $P_{Edem1}$ = 0.8258, ns $P_{Atf4}$ = 0.1330, ns $P_{Chop}$ = 0.7046, ns $P_{Gadd34}$ = 0.6283, ns $P_{Atf6}$ = 0.8739, ns $P_{Calr}$ = 0.8541). Data information: results are shown as mean ± SD and significance was calculated by unpaired two-tailed *t* test.
Source data are available for this figure.

Norton et al, 2022), these cells may cope with continued and constitutive progerin expression by down-regulating these ER stress response genes back to basal expression levels to maintain proteostasis and escape apoptosis. Non-endothelial cells, on the other hand, particularly VSMCs, may not be able to adapt to continuous stress and maintain a dysregulated activation of the UPR pathways, resulting in apoptosis induction (Hamczyk et al, 2019). However, further studies will be necessary to test this hypothesis.

Overall, our data shed new light on the contributions of arterial cell types to CVD in HGPS. While up-regulation of the UPR in VSMCs contributes to VSMC loss by apoptosis, eventually leading to accumulation of lipids in the media and development of atherosclerosis (Hamczyk et al, 2018, 2019), ECs may contribute to CVD by other means. Based on the phenotypes reported for ECs in different HGPS mouse models and in iPSC-derived HGPS EC models (Benedicto et al, 2025a), their contributions to CVD may be mediated by their reduced nitric oxide (NO) production (Matrone et al, 2019; Gete et al, 2021; Vakili et al, 2025), the development of a senescent phenotype and initiation of a pro-inflammatory and pro-fibrotic response (Matrone et al, 2019; Osmanagic-Myers et al, 2019; Bidault et al, 2020; Sun et al, 2020; Mojiri et al, 2021; Manakanatas et al, 2022; Xu et al, 2022; Barettino et al, 2024), defective angiogenesis (Gete et al, 2021;

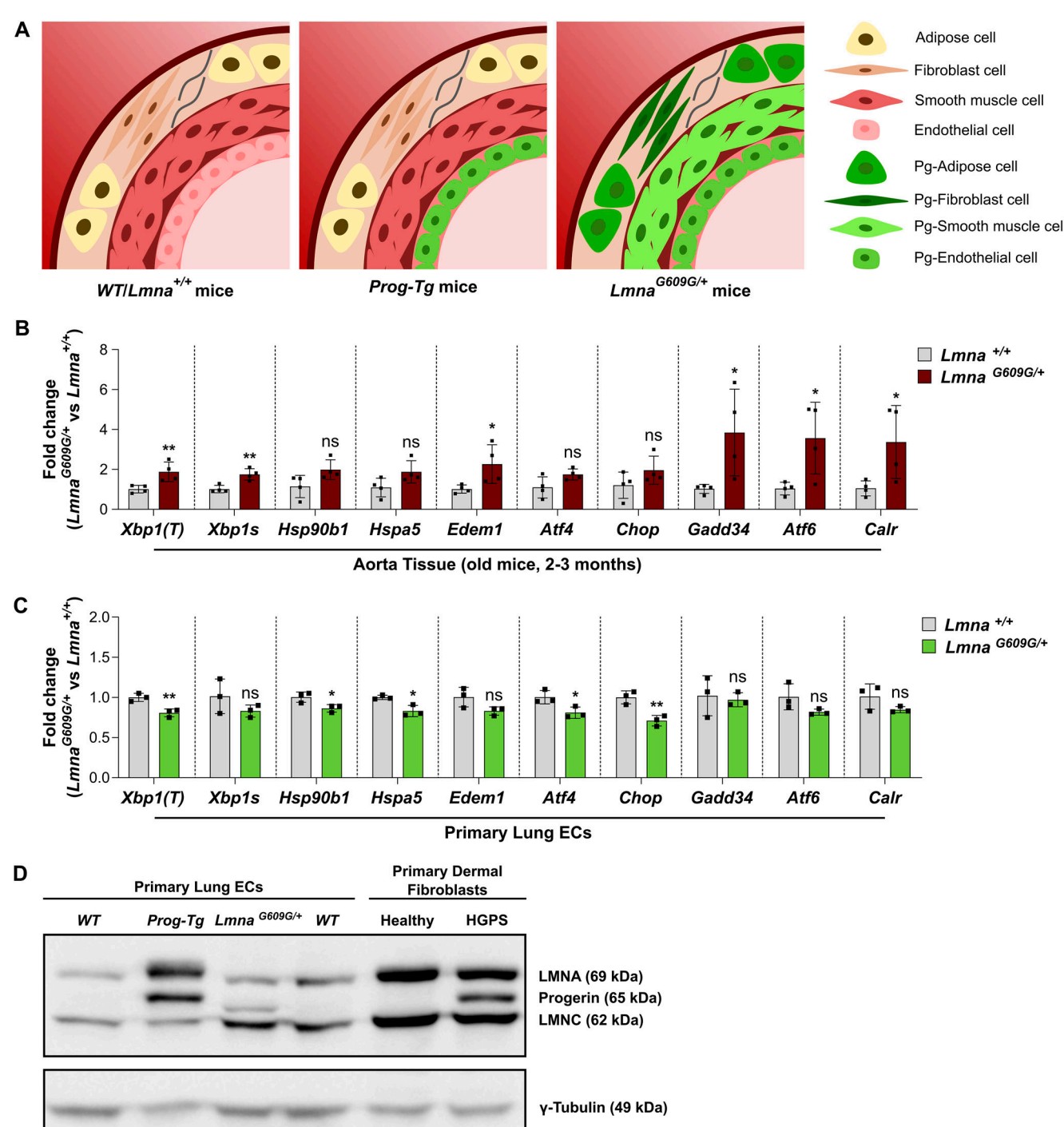

**Figure 7. Activation of the unfolded protein response in the aorta of *Lmna^G609G/+* mice originates from non-endothelial cells.**
**(A)** Representation of a cross-section of the arterial wall from *WT*/*Lmna^+/+* mice **(left)**, *Prog-Tg* mice **(middle)**, and *Lmna^G609G/+* mice **(right)**. The arterial wall is made up of three distinct layers: tunica intima, with endothelial cells; tunica media with vascular smooth muscle cells; and tunica adventitia, with fibroblasts and adipose tissue cells. Whereas *Prog-Tg* mice express progerin exclusively in endothelial cells, *Lmna^G609G/+* mice have ubiquitous progerin expression in all cell types. Progerin-expressing cells are shown in green. **(B)** qRT-PCR analysis of *Xbp1(T)*, *Xbp1s*, *Hsp90b1*, *Hspa5*, *Edem1*, *Atf4*, *Chop*, *Gadd34*, *Atf6*, and *Calr* expression in whole aorta tissue lysates from older *Lmna^+/+* and *Lmna^G609G/+* mice. Expression levels were normalized to the reference gene *Hprt* and to *Lmna^+/+* samples (n = 4 mice) (**$P_{Xbp1(T)}$ = 0.0089, **$P_{Xbp1s}$ = 0.0035, ns $P_{Hsp90b1}$ = 0.1095, ns $P_{Hspa5}$ = 0.0827, *$P_{Edem1}$ = 0.0204, ns $P_{Atf4}$ = 0.0895, ns $P_{Chop}$ = 0.1981, *$P_{Gadd34}$ = 0.0129, *$P_{Atf6}$ = 0.0214, *$P_{Calr}$ = 0.0259). **(C)** qRT-PCR analysis of *Xbp1(T)*, *Xbp1s*, *Hsp90b1*, *Hspa5*, *Edem1*, *Atf4*, *Chop*, *Gadd34*, *Atf6*, and *Calr* expression in primary lung ECs isolated from *Lmna^+/+* and *Lmna^G609G/+* mice. Expression levels were normalized to the reference gene *Hprt* and to *Lmna^+/+* samples (n = 3 biological replicates) (**$P_{Xbp1(T)}$ = 0.0099, ns $P_{Xbp1s}$ = 0.2308, *$P_{Hsp90b1}$ = 0.0403, *$P_{Hspa5}$ = 0.0191, ns $P_{Edem1}$ = 0.0848, *$P_{Atf4}$ = 0.0350, **$P_{Chop}$ = 0.0084, ns $P_{Gadd34}$ = 0.8456, ns $P_{Atf6}$ = 0.1026, ns $P_{Calr}$ = 0.1589). **(D)** Immunoblot analysis of *WT*, *Prog-Tg*, and *Lmna^G609G/+* lung EC lysates, stained with an anti-LMNA/C (E1) antibody **(top)** and an anti-γ-Tubulin antibody **(bottom)**. A Hutchinson-Gilford Progeria Syndrome fibroblast patient sample, together with a healthy control, was used to assess antibody specificity and identify progerin. Note that *Prog-Tg* and Hutchinson-Gilford Progeria Syndrome fibroblast samples express human progerin, whereas *Lmna^G609G/+* mice express endogenous mouse progerin. **(B, C)** Data information: in (B, C),

Vakili et al, 2025), altered mechanosensing (Osmanagic-Myers et al, 2019; Atchison et al, 2020; Danielsson et al, 2020, 2022; Barettino et al, 2024), and an endothelial-to-mesenchymal transition (Hamczyk et al, 2024). Thus, our data have strong implications for future therapies targeting proteostasis that should discern the effects on non-endothelial cells, mainly VSMCs, versus ECs. Several strategies to ameliorate proteostasis were shown to delay aging and the development of multiple age-related diseases (Lopez-Otin et al, 2023). These include recombinant and chemical chaperones to improve protein folding capacity (Bobkova et al, 2015; Hafycz et al, 2022), chaperone-mediated autophagy (CMA) inducers to remove misfolded and damaged proteins (Bourdenx et al, 2021; Dong et al, 2021; Madrigal-Matute et al, 2022), and guanabenz, an antihypertensive and proposed modulator of eIF2α phosphorylation (Dalla Bella et al, 2021). The chemical chaperone tauroursodeoxycholic acid (TUDCA) is particularly promising, as it has been shown to alleviate ER stress and reduce apoptosis in non-endothelial cells from different HGPS models (Hamczyk et al, 2019; Vidak et al, 2023).

While these proteostasis-linked treatments may be effective in non-endothelial cells, further investigations on the molecular mechanisms that drive EC dysfunction are still needed to develop new strategies to prevent or reverse EC damage and, consequently, prevent CVD.

# Materials and Methods

### Mice

Bi-transgenic *Prog-Tg* mice were generated as previously described (Osmanagic-Myers et al, 2019). Mice expressing a tet operon-driven HGPS mutation (1824C>T, G608G, *tetop-LA$^{G608G}$*, C57BL/6J background) were crossed with transgenic mice carrying a tetracycline-responsive transcriptional activator under the control of the EC-specific VE-Cadherin promoter (*Cdh5-tTA* mice, MGI:4437711, FVB background; Jackson Laboratories). Mice were kept without doxycycline, allowing constitutive expression of the mutant human Lamin A minigene (LA$^{G608G}$) exclusively in ECs. Alternatively, for acute progerin expression analysis, doxycycline hyclate (10 mg/ml, D9891; Sigma-Aldrich) was added to drinking water of mice, supplemented with saccharose (2.5%, 4,661.1; Roth), to prevent progerin expression. Genotyping was performed with DNA extracted from the toe using the primers described in Table 1.

Generation of the *Lmna$^{G609G}$* knock-in mouse model was previously described (Osorio et al, 2011). These mice carry the HGPS mutation (c.1827C>T;p.Gly609Gly), which is equivalent to the human HGPS mutation (c.1824C>T;p.Gly608Gly).

Both male and female mice were used in the study, but gender-specific effects were not considered due to animal availability.

### Whole tissue isolation

*WT* and *Prog-Tg* littermates were euthanized at an age of 6 mo for heart isolation and at 3.5 mo for aorta isolation. *Lmna$^{+/+}$* and *Lmna$^{G609G/+}$* mice were euthanized at 2–3 mo for aorta isolation. Briefly, mice were euthanized and the organs were perfused with cold PBS (D8537; Sigma-Aldrich) to remove any remaining blood. The heart was then isolated, sliced into thinner sections, and transferred to an RNAprotect Tissue Reagent (76104; QIAGEN), according to manufacturer instructions. Aorta samples were first cleaned of the adventitia layer before being stored in the RNAprotect Reagent. Afterward, tissues were transferred to QIAzol lysis reagent (79306; QIAGEN) and further homogenized using 2.8 mm Precellys zirconium oxide ceramic beads (Kit CK28; Bertin) and a Precellys 24 tissue homogenizer (Bertin Instruments). The samples were then used for RNA extraction, as described below.

### Heart EC isolation

*WT* and *Prog-Tg* littermates were euthanized at 6 mo for heart EC isolation. Briefly, the hearts were removed and minced with scalpels, followed by incubation with 200 U/ml collagenase type I (17100017; Gibco) for 45 min at 37°C, with rotation. The collagenase and tissue solution were then passed through an 18-gauge needle and syringe, and filtered using a 70 μm cell strainer (431751; Corning). Cells were then centrifuged at 250*g* for 5 min and resuspended in 1 ml of cold culture medium. The cell suspension was incubated for 25 min at 4°C, on end-over-end rotation, with 4 μl of ICAM-2 (553326; BD Biosciences)-coupled magnetic Dynabeads (11035; Invitrogen). The samples were then put on a magnetic stand and washed two times with PBS supplemented with 0.1% BSA (P06-139210; PAN-Biotech) and 2 mM EDTA (15575-038; Invitrogen). After the last wash, the ECs that were present in the magnetic pellet were directly resuspended in QIAzol lysis reagent (79306; QIAGEN) and used for RNA extraction, as described below.

### Lung cell isolation and sorting into ECs and non-ECs

Primary cells were isolated from mice, using a previously established protocol (Osmanagic-Myers et al, 2019). Briefly, the lungs were isolated from 8 to 21 d-old littermates and pooled together (1–4 lungs per biological replicate). The tissue was then minced with scalpels and incubated with 200 U/ml collagenase type I (17100017; Gibco) for 45 min at 37°C, on end-over-end rotation. Afterward, the digested tissue was passed 15 times through an 18-gauge needle and syringe, and filtered through a 70 μm cell strainer (431751; Corning). Cells were then centrifuged at 250*g* for 8 min and further plated on 0.8% wt/vol gelatin- (G1393; Sigma-Aldrich) and 10 μg/ml fibronectin-coated (1918-FN; R&D Systems) dishes. The following day, plates were cleaned with PBS to remove dead cells. After reaching confluency, typically 48 h after isolation, cells were detached using trypsin-EDTA (T4049; Sigma-Aldrich) for

---

results are shown as mean ± SD. In (B), significance was calculated after batch correction by unpaired two-tailed *t* test. In (C), significance was calculated by unpaired two-tailed *t* test.
Source data are available for this figure.

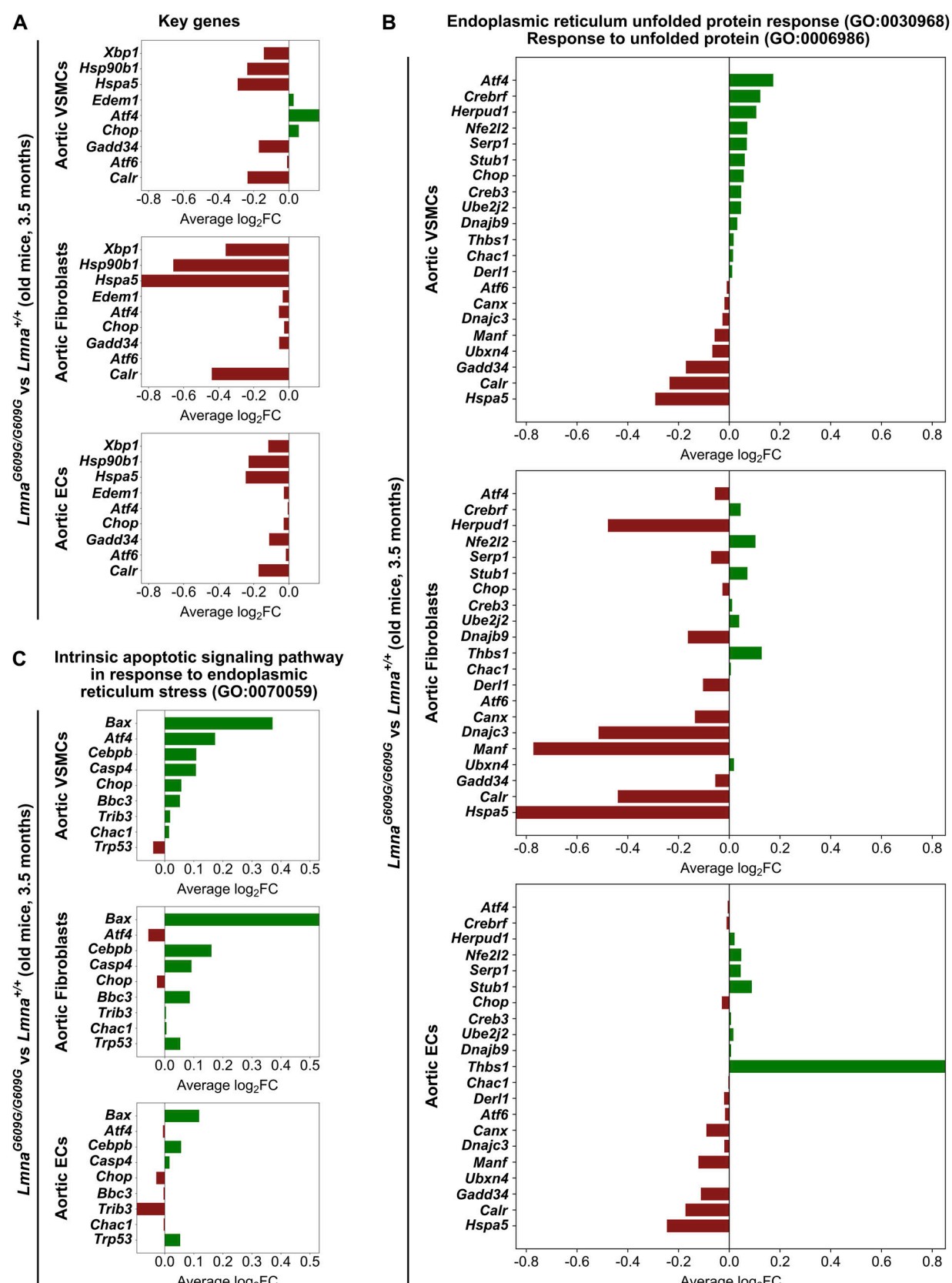

**Figure 8. Dysregulated unfolded protein response activation is exclusive to vascular smooth muscle cells (VSMCs) in arteries of *Lmna^{G609G/G609G}* mice.**
**(A)** Differential expression analysis of *Xbp1*, *Hsp90b1*, *Hspa5*, *Edem1*, *Atf4*, *Chop*, *Gadd34*, *Atf6*, and *Calr* in combined aortic VSMC-, fibroblast-, and EC-subpopulations, based on the scRNA-Seq dataset from whole aorta of *Lmna^{G609G/G609G}* versus *Lmna^{+/+}* mice (Barettino et al, 2024). **(B, C)** Analysis of differentially expressed genes

3 min at 37°C, centrifuged at 250$g$ for 4 min, and resuspended in 1 ml of cold culture medium. The cell suspension was then incubated for 30–45 min at 4°C, on end-over-end rotation, with 5–20 $\mu$l of magnetic Dynabeads (11035; Invitrogen) that had been previously coupled to an ICAM-2 antibody (553326; BD Biosciences). This allowed for the separation and sorting of the different cell types using a magnetic stand and five consecutive washes in PBS supplemented with 0.1% BSA (P06-139210; PAN-Biotech) and 2 mM EDTA (15575-038; Invitrogen). Non-ECs were recovered from the supernatant from the first wash and plated on non-coated dishes. On the other hand, ECs bound to the ICAM-2-coupled Dynabeads were resuspended in culture medium after five washes, and then plated on coated dishes. All primary cells (endothelial and non-endothelial) were used for experiments after no more than three passages after isolation.

## Cell culture and treatments

Primary HGPS dermal fibroblast cell lines were obtained from the Coriell Cell Repository (CCR). An HGPS cell line (AG11513, female, 8 yr old) and a healthy control (GM00323, male, 11 yr old) were used. Cells were cultured in Dulbecco's modified Eagle's medium (DMEM, D5796; Sigma-Aldrich) supplemented with 15% fetal bovine serum (FBS, F7524; Sigma-Aldrich), 100 U/ml penicillin, 100 $\mu$g/ml streptomycin (P4333; Sigma-Aldrich), and 2 mM L-glutamine (P04- 80100; PAN-Biotech), and grown in a 37°C humidified incubator with 5% $CO_2$.

Endothelial and non-endothelial cells were cultured in DMEM (D5796; Sigma-Aldrich) supplemented with 20% FBS (F7524; Sigma-Aldrich), EC growth supplement (PB-1166; CellBiologics), 25 mM HEPES (H0887; Sigma-Aldrich), 100 U/ml penicillin, 100 $\mu$g/ml streptomycin (P4333; Sigma-Aldrich), 2 mM L-glutamine (P04-80100; PAN-Biotech), non-essential amino acids (P08-32100; PAN-Biotech), 1 mM sodium pyruvate (11360-070; Gibco), and 100 $\mu$g/ml heparin (H3149; Sigma-Aldrich), and grown in a 37°C humified incubator with 5% $CO_2$ and 8% $O_2$.

To induce ER stress in ECs, Tunicamycin (2.5 $\mu$g/ml, T7765; Sigma-Aldrich) was added to the culture medium. Cells were treated for 2, 4, 8, and 24 h at 37°C to determine optimal treatment duration. The 8 h timepoint was chosen for remaining experiments, and a DMSO (2.5 $\mu$g/ml, D2650; Sigma-Aldrich) control was used for untreated conditions.

To temporally control progerin expression in ECs, doxycycline hyclate (10 mg/ml, D9891; Sigma-Aldrich) was added to the culture medium of cells isolated from mice with doxycycline drinking water (see above). After cell sorting, doxycycline was removed from the culture medium at different timepoints (2d, 5d, 8d, and 15d), to induce progerin expression. A control that was kept in doxycycline was included in the experiments.

## RNA isolation and gene expression analysis

RNA was isolated from tissues (heart and aorta) and primary cells with the miRNeasy Mini Kit (217004; QIAGEN), according to the manufacturer's protocol, and then quantified using a spectrophotometer (DeNovix DS-11FX). To analyze gene expression, cDNA was transcribed using a cDNA Synthesis Kit (A3500; Promega). qRT-PCR was then performed using the KAPA SYBR FAST Universal (KK4618; Sigma-Aldrich) on a CFX96 Real-Time PCR Detection System (Bio-Rad). Reaction conditions were: 3 min at 95°C, 1 cycle; 10 s at 95°C, 40 s at 60°C, 40 cycles. All reactions were performed in triplicate and normalized to the internal control hypoxanthine-guanine phosphoribosyl-transferase (*Hprt*). Analyzed primer pairs are listed in Table 1. Gene expression analysis was performed based on the ΔΔCt method. Ct values varying more than 0.5 between technical replicates were excluded from analysis, and only the two replicates closest in Ct value were used. Statistical analysis was performed using the ΔCt values (or corrected ΔCt values when batch correction was performed).

## Immunofluorescence staining

For immunofluorescence analysis, cells were plated in $\mu$-Slide eight-well ibidi chambers (80826; ibidi) and fixed with 4% PFA, for 10 min, at RT. This was followed by two washes in PBS, and permeabilization in 0.1% Triton-X/PBS for 6 min. Samples were washed again and blocked in 3% BSA in 0.1% Tween/PBS for 1 h at RT. Cells were then incubated with the primary antibodies overnight at 4°C: rabbit anti-ATF4 monoclonal (1:200, 11815; Cell Signaling), rat anti-CD31 monoclonal (1:100, 553370; BD Biosciences), mouse anti-LMNA/C (JOL2) monoclonal (1:30, ab40567; Abcam), rabbit anti-XBP1s monoclonal (1:400, 40435; Cell Signaling). The following day, cells were washed with 0.05% Tween/PBS and stained for 1 h at RT with the secondary antibodies, as follows: donkey anti-mouse DyLight 488 (1:200, SA5-10166; Thermo Fisher Scientific), donkey anti-rat DyLight 594 (1:200, SA5-10028; Thermo Fisher Scientific), goat anti-rabbit DyLight 594 (1:200, 35561; Thermo Fisher Scientific). This was followed by more washing steps with 0.05% Tween/PBS, and counterstaining with Hoechst (1:10,000 in PBS, 62249; Thermo Fisher Scientific) for 5 min at RT. Mounting medium (50001; ibidi) was then added to the chambers.

## Light microscopy and image quantification

Imaging was performed on an inverse confocal microscope (Zeiss LSM 980) equipped with EC Plan-Neofluar 10x/0.3, WD 5.2 mm and Plan-Apochromat 40x/1.4 Oil DIC, WD 0.13 mm objectives. All images were acquired using Zeiss ZEN 3.3 Software, with the same exposure settings for all samples. Raw unprocessed data were then used for further quantification with ImageJ/Fiji Software. Representative images were equally adjusted for brightness and contrast and exported to Inkscape.

Quantification of nuclear intensity for proteins of interest was carried out using a macro created with the image analysis software ImageJ/Fiji. Using the Hoechst channel, noise was removed using

previously identified in a dysfunctional *Lmna*$^{G609G/G609G}$ VSMC subpopulation (Barettino et al, 2024), belonging to the GO terms "Endoplasmic reticulum unfolded protein response" and "Response to unfolded protein" (B), and GO term "Intrinsic apoptotic signaling pathway in response to endoplasmic reticulum stress" (C), in combined aortic VSMC-, fibroblast-, and EC-subpopulations in *Lmna*$^{G609G/G609G}$ versus *Lmna*$^{+/+}$ samples (Barettino et al, 2024). Up-regulated genes are represented in green, and down-regulated genes are shown in red. The X-axis indicates the average log$_2$ values of fold change.

**Table 1.   List of primer sequences**

| Target | Primer forward (5′-3′) | Primer reverse (5′-3′) |
|---|---|---|
| qRT-PCR | | |
| Atf4 | GGGTTCTGTCTTCCACTCCA | AAGCAGCAGAGTCAGGCTTTC |
| Atf6 | GCGGATGATAAAGAACCGAGAG | ACAGACAGCTCTTCGCTTTG |
| Calr | GCTACGTGAAGCTGTTTCCGA | ACATGAACCTTCTTGGTGCCAG |
| Chop | GGAGCTGGAAGCCTGGTATG | GGATGTGCGTGTGACCTCTG |
| Edem1 | CTACCTGCGAAGAGGCCG | GTTCATGAGCTGCCCACTGA |
| Gadd34 | CTTTTGGCAACCAGAACCG | CAGAGCCGCAGCTTCTATCT |
| Hprt | GCAGTCCCAGCGTCGTGATTA | TGATGGCCTCCCATCTCCTTCA |
| Hsp90b1 | AAGAATGAAGGAAAAACAGGACAAAA | CAAATGGAGAAGATTCCGCC |
| Hspa5 | ACCCACCAAGAAGTCTCAGATCTT | CGTTCACCTTCATAGACCTTGATTG |
| Pecam | GGAAGCCAACAGCCATTACG | TCCGTTCTCTTGGTGAGGCT |
| Progerin | ACTGCAGCAGCTCGGGG | TCTGGGGGCTCTGGGC |
| Xbp1(T) | ACATCTTCCCATGGACTCTG | TAGGTCCTTCTGGGTAGACC |
| Xbp1s | GAGTCCGCAGCAGGTG | TCCAGAATGCCCAAAAGG |
| Genotyping | | |
| Cdh5-tTA | CGCTGTGGGGCATTTTACTTTAG | CATGTCCAGATCGAAATCGTC |
| Mutant human LMNA minigene (LA$^{G608G}$) | GCAACAAGTCCAATGAGGACCA | GTCCCAGATTACATGATGC |

the "Smooth" function; threshold was adjusted using the "Li" setting, giving an image with a black background and with nuclei portrayed in white; any holes within the nuclei were filled using "Fill Holes"; nuclei edges were marked using "Find Edges." Then, after automatically switching to the channel containing the protein of interest, nuclei edges from Hoechst channel were opened on ROI manager; "Integrated Density (IntDen)" and "Area" were obtained using "Analyse Particles," excluding any nuclei touching the edges of the field and only analyzing nuclei with a size varying from 10,000-infinite pixel-square. Corrected nuclear intensity for the protein of interest was then calculated using the previously described formula (Gavet & Pines, 2010):

$$Corrected\ intensity = Integrated\ Density - (Area\ of\ cell \times fluorescence\ of\ Background)$$

### Immunoblotting

For protein extraction, confluent cells were trypsinized and washed with PBS, before being pelleted by centrifugation at 9,500$g$ for 1 min at 4°C. The pellet was then resuspended in sample buffer (50 mM Tris–HCl, 100 mM DTT, 2% SDS, 0.1% bromophenol blue and 10% glycerol pH 6.8), triturated with an 18-gauge needle, and incubated for 5 min at 95°C. Protein lysates were separated on SDS polyacrylamide gels and transferred onto nitrocellulose membranes (10600001; Cytiva) at 25 V, 4°C, overnight. The following day, protein transfer was confirmed by incubation with Ponceau Solution (161470100; Thermo Fisher Scientific), and blocking was performed in 5% milk in PBS-T for 1 h at RT, followed by incubation with the primary antibodies overnight at 4°C: mouse anti-LMNA/C (E1) monoclonal (1:1,000, sc-376248;

Santa Cruz Biotechnology), mouse anti-γ-Tubulin monoclonal (1:5,000, T6557; Sigma-Aldrich). On the next day, membranes were incubated with HRP-coupled secondary antibody for 1 h at RT: donkey anti-mouse HRP (1:15,000, 715-035-151; Jackson ImmunoResearch). Bands were then visualized using the SuperSignal West Pico PLUS Chemi-luminescent Substrate (34580; Thermo Fisher Scientific) and the ChemiDoc Touch Imaging System (Bio-Rad). Image analysis was performed with the Image Lab Software (Bio-Rad).

### Flow cytometry

For intracellular stainings, cells were collected from cell culture plates by trypsinization, and resuspended in PBS. Cells were washed twice by centrifugation at 270$g$ for 4 min, and then resuspended and incubated in 4% PFA for 15 min at RT for fixation. Afterward, the samples were centrifuged at 900$g$ for 5 min and then washed in FACS Buffer (0.01% BSA, 5 mM EDTA pH 8, PBS) at 900$g$ for 5 min. This was followed by permeabilization with ice-cold methanol for 10 min, on ice. After incubation, FACS Buffer was added to the samples to stop permeabilization, and methanol was removed by centrifugation at 900$g$ for 10 min. The samples were further washed with FACS Buffer, and incubated with primary antibodies for 1 h at RT: rabbit anti-ATF4 monoclonal (1:100, 11815; Cell Signaling), rabbit anti-XBP1s monoclonal (1:200, 40435; Cell Signaling). Additional washing steps were performed with FACS Buffer before staining with secondary antibodies for 15 min, on ice: goat anti-rabbit DyLight 594 (1:200, 35561; Thermo Fisher Scientific). Final washing steps were performed, and the samples were then resuspended in FACS Buffer and transferred to FACS tubes (352235; Corning).

For GFP signal detection, cells were collected and directly transferred to FACS tubes for analysis, without additional staining.

All samples were analyzed on a Cell Analyzer (BD LSRFortessa) with the FACSDiva Software (BD Biosciences), using the 488-nm laser for green signal, and the 561-nm laser for red signal detection. Data analysis was then performed using the FlowJo Software (BD Biosciences).

### scRNA-seq data processing, differential expression analysis, and visualization

Publicly available single-cell RNA-Seq datasets of the proximal aorta from $Lmna^{G609G/G609G}$ progeroid mice and $Lmna^{+/+}$ controls were used for the analysis (Barettino et al, 2024). 10x Genomics Cell Ranger outputs (count matrices, barcodes, and feature annotation files) were obtained from ArrayExpress (E-MTAB-13678) (Parkinson et al, 2007) and imported into Python using the Scanpy framework (v1.9) (Wolf et al, 2018).

Four independent samples (two $Lmna^{+/+}$ and two $Lmna^{G609G/G609G}$) were combined into a single AnnData object, whereas retaining per-sample identifiers to allow separation by genotype in subsequent analyses. Cell-level metadata provided by the original study (Barettino et al, 2024), including cluster assignments, genotype, and sample information, was merged into this object. Gene identifiers were linked to gene symbols using Ensembl v84 annotations. After processing and metadata integration, the dataset comprised 31,760 cells and 28,692 genes.

Differential expression analysis focused on three major vascular cell populations—VSMCs, fibroblasts, and ECs. Previously defined subclusters of each cell type (Barettino et al, 2024) were merged into broader lineage-level groups to increase statistical power and facilitate comparisons across the cell populations of interest (VSMCs: FVC1 + FVC3 + FVC4 + FVC7; fibroblasts: FVC0 + FVC2 + FVC5 + FVC8; ECs: EC0 + EC1 + EC2 + EC3 + EC4 + EC5 + EC6 + EC8 + EC9; see [Barettino et al, 2024] for original subclustering strategy). Merging was performed in Python using the Scanpy framework. Each cell in the AnnData object carried a metadata column that assigned it to an author-defined subcluster. Dictionaries mapping these subcluster identifiers to higher level groups were then created. For each group, cells were further subsetted according to these mappings, combined into a merged population, and differential expression analysis was performed on the aggregated set. Within each merged group, cells were then further stratified by genotype ($Lmna^{G609G/G609G}$ versus $Lmna^{+/+}$). For each comparison, raw counts were normalized to a library size of 10,000 transcripts per cell and log-transformed (natural log). Differential expression was then computed in a Seurat-like manner (Stuart et al, 2019), with average log-normalized expression values calculated for each genotype, and their differences reported as $\log_2$ values of fold change. Differential expression results are presented as descriptive comparisons of average $\log_2$-transformed expression patterns without statistical hypothesis testing.

Plotting was performed in Python using Matplotlib (Hunter, 2007). Bars were colored green for genes up-regulated in $Lmna^{G609G/G609G}$, or red for genes down-regulated in $Lmna^{G609G/G609G}$. Consistent x-axis scaling across cell types was used for direct comparison between VSMCs, fibroblasts, and ECs. In Fig 8B and C, the genes were sorted in all plots based on the order of average $\log_2$FC values in aortic VSMCs.

### Statistical analysis

All data were analyzed using Graph Prism Software 8.0. The number of analyzed mice, cells, or biological replicates for each experiment is specified in the figure legends as n. Normal distribution of all data was assessed with the D'Agostino-Pearson and Shapiro-Wilk normality tests. Batch correction was additionally performed for qRT-PCR analysis when data were generated across multiple experimental runs to account for variability in Ct values and is indicated in the respective figure legends. Statistical significance was calculated using unpaired two-tailed $t$ test, repeated-measures one-way ANOVA test, or repeated-measures Kruskal-Wallis test, as specified in the figure legends. The following $P$-value ranges were used for statistical significance for all statistical tests: ns ($P > 0.05$); * ($P \leq 0.05$); ** ($P \leq 0.01$); *** ($P \leq 0.001$); **** ($P \leq 0.0001$).

### Study approval

All mouse experiments were approved (No: 2021-0.873.416; No: 2020-0.469.732; July 2020; amendment 2024-0.766.661) by the regional Ethics Committee for Laboratory Animal Experiments at the Medical University of Vienna and the Austrian Ministry of Science Research and Economy (BMWFW-66.009/0321-WF/V/3b/2016 and BMWFW-66.009/0156-WF/V/3b/2017), according to Austrian Law BGBl. I Nr.114/2012 (TVG2012) idF BGB I Nr.76/2020 and in accordance with the Guide for the Care and Use of Laboratory Animals published by the US National Institutes of Health (NIH Publication No. 85-23, revised 1996).

# Data Availability

This study includes no data deposited in external repositories. All raw data are included in the article as source data for the corresponding Figure.

# Supplementary Information

# Acknowledgements

This research was funded in whole or in part by the Austrian Science Fund (FWF) (P 37003) to S Osmanagic-Myers and (I 4694-B) to R Foisner, the latter under the frame of EJP RD, the European Joint Programme on Rare Diseases. In addition, this project has received funding from the European Union's Horizon 2020 research and innovation programme under the EJP RD COFUND-EJP N 825575, and a doctorate program funded by the Austrian Science Fund (FWF) (W1261-B28). For open access purposes, the author has applied a CC BY public copyright license to any author accepted manuscript version arising from this submission. RA Silva is a recipient of a DOC Fellowship of the Austrian Academy of Sciences at the Max Perutz Labs, Medical

University Vienna (Ö ÖAW DOC 26995). We thank Maria Eriksson (Karolinska Institutet, Sweden) and Carlos López-Otín (Universidad de Oviedo, Spain) for providing *Lmna*^*G609G/+* mice. The authors thank Prof. Dr. Giovanna Lattanzi (IGM-CNR, Bologna) for valuable discussions and collaborative work for this project. The authors would like to acknowledge the BioOptics facility at the Max Perutz Labs, where microscopic and FACS analyses were performed, as well as the animal facility at Max Perutz Labs. We also thank Filip Milosic (Medical University of Vienna) for helping in the isolation and preparation of the *Lmna*^*+/+* and *Lmna*^*G609G/+* mouse samples. Lastly, we thank Petra Fichtinger and Irene Gösler (both Medical University Vienna) for their assistance with genotyping.

## Author Contributions

RA Silva: conceptualization, data curation, formal analysis, funding acquisition, investigation, visualization, methodology, and writing—original draft, review, and editing.
F Sarigol: software, formal analysis, and writing—review and editing.
GE Karagöz: conceptualization, supervision, and writing—review and editing.
S Osmanagic-Myers: conceptualization, supervision, funding acquisition, and writing—review and editing.
R Foisner: conceptualization, supervision, funding acquisition, project administration, and writing—review and editing.

## Conflict of Interest Statement

The authors declare that they have no conflict of interest.

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
