## [Reviewer comments · Life Science Alliance]

The unfolded protein response in progeria arteries originates from non-endothelial cell types

Raquel Silva, Fatih Sarigol, Elif Karagöz, Selma Osmanagic-Myers, and Roland Foisner
DOI: <https://doi.org/10.26508/lsa.202503485>

Corresponding author(s): Roland Foisner, Medical University of Vienna and Selma Osmanagic-Myers, Medical University of Vienna

Review Timeline:

Submission Date:	2025-08-13
Editorial Decision:	2025-08-19
Revision Received:	2025-11-03
Editorial Decision:	2025-11-06
Revision Received:	2025-11-11
Accepted:	2025-11-17

Scientific Editor: Tim Fessenden

Transaction Report:

Please note that the manuscript was previously reviewed at another journal and the reports were taken into account in the decision-making process at *Life Science Alliance*.

Referee #1 Review

Report for Author:

This is a manuscript by Silva et al exploring the molecular pathogenesis of Hutchinson-Gilford progeria syndrome (HGPS), which is a rare, premature aging disorder. One of the hallmarks of HGPS is accelerated cardiovascular disease, in a form of severe atherosclerosis that underlies arterial stiffening as well as myocardial infarction if left untreated. As prior studies have linked dysregulated activation of unfolded protein response (UPR) in vascular dysfunction, the authors focused specifically on whether progerin expression in endothelial cells (ECs) can elicit ER stress directly. Using progerin-expressing ECs from an EC specific progeria mouse model (Prog-Tg), the authors investigated the three canonical UPR pathways. They found that constitutive progerin expression in ECs alone does not activate UPR despite retaining the ability to do so. Furthermore, they show that arteries from mice ubiquitously expressing progerin display UPR activation but not in EC-specific expression, demonstrating that UPR in HGPS originates from non-endothelial arterial cell populations.

The strength of the manuscript is that the authors rigorously examined previously identified disease mechanisms in a cell type specific-manner in the various cells that form the vasculature. This has led to a deeper understanding of how vascular complications arise in HGPS - that dysregulated UPR is not directly linked to progerin expression in ECs and likely occurs other cell types of the vasculature. A major weakness is that it does not push the boundaries of our current understanding. No novel pathways or molecular pathogenesis has been linked to HGPS. Despite this, the study findings are important, timely, and provide sufficient conceptual advance critical for understanding the pathogenesis of cardiovascular complications in HGPS.

Points to consider:

- 1) It's not clear to be what's the purpose of presenting data in Fig, EV1C and EV1D. The text descriptor states "it is important to note that these samples had lower progerin expression than the cultured ECs, given that non-ECs in these tissues do not express progerin" but the data shows neither of these.
- 2) Why use ECs from the lungs? Wouldn't it be more appropriate to isolate ECs from myocardial vasculature? Some explanation would be helpful.
- 3) It's a missed opportunity that the lung wasn't stressed in any way to ascertain whether any sort of physiological stress can activate UPR in progerin expressing ECs.
- 4) Acute progerin expression leading to XBP1 splicing is interesting. Although the use of cells from in vivo model to demonstrate this effect is commendable, it is unclear whether the nuclear enrichment of XBP1s is due to progerin expression itself or due to stress inherent to any transgene overexpression.
- 5) The predominant localization of GFP in ECs from Pro-Tg mice is somewhat surprising. Typically, GFP without any localization signal (expressed via IRES) would look like diffuse GFP staining the whole cell. However, authors see distinct nuclear rim staining, reminiscent of lamin A-GFP fusion protein. Some explanation would be helpful.
- 6) Although I appreciate the effort to perform kinetic analysis (Fig. EV2) to determine the optimal timepoint for maximal UPR gene expression (Fig. 4A-C), the approach taken may miss potentially altered expression kinetics caused by progerin.

Referee #2 Review

Report for Author:

Silva and colleagues investigated the effects of progerin in endothelial cells (ECs) on the unfolded protein response (UPR) in the context of Hutchinson-Gilford progeria syndrome (HGPS), a premature aging disorder characterized by accelerated cardiovascular disease (CVD). While previous studies have implicated UPR activation in vascular smooth

muscle cells (VSMCs) and fibroblasts, the contribution of ECs to this process remains unclear.

The authors used a transgenic mouse model expressing progerin specifically in ECs (Prog-Tg) and assessed UPR activation through canonical ER stress sensors (IRE1 α , PERK, and ATF6). They found that sustained progerin expression in ECs does not induce a robust UPR at baseline, although the cells remain responsive to tunicamycin-induced ER stress. Interestingly, acute progerin expression does activate UPR through the IRE1 α pathway, suggesting a potential adaptive mechanism that resets UPR signaling to basal levels under chronic progerin exposure. The investigations were extended to in vivo models. They confirmed earlier studies showing UPR activation in aortas of Lmna^{G609G} mice (that express progerin in all cells) but they did not observe UPR activation in Prog-Tg mice, supporting the idea that UPR activation in HGPS arteries originates from non-endothelial cells.

The manuscript is clearly written, and the data are well presented. However, several points should be addressed to strengthen the conclusions.

Comments:

1. The authors should clarify the rationale for using lung ECs rather than ECs from progeria-affected arteries (e.g., aorta). The strategy to isolate ECs (ICAM-2 expression) likely yields predominantly capillary ECs. Are these representative of arterial ECs in terms of gene expression, function, or disease relevance in HGPS? The authors should discuss whether their findings can be generalized to arterial ECs.

2. ICAM-2 is expressed by multiple EC subtypes (capillary, arterial, venous, and possibly lymphatic). The composition of the lung EC preparation should be better characterized. The authors should clarify the relative proportions of these subtypes and whether progerin expression alters this composition—for example, by reducing the proportion of arterial ECs.

3. The conclusion that UPR activation is cell type-specific assumes comparable progerin expression in ECs across the Lmna^{G609G} and Prog-Tg models. This assumption should be experimentally validated. Ideally, progerin levels should be measured in aortic ECs. If this is not feasible, Western blotting of lung ECs using an antibody that cross-reacts with both human and mouse progerin should be performed.

4. The hypothesis that constitutive progerin expression induces an adaptive mechanism to reduce baseline UPR activation is novel and intriguing. To validate this idea, control studies with ECs isolated from control LA-Tg mice (that express GFP and wild type lamin A) should be included. The authors have used this model in previous studies (Osmanagic-Myers et al., 2019).

Minor Comments:

1. The claim that primary lung ECs are highly pure is not fully supported by Figure 1F, where approximately 45% of cells in the image (5/11) appear to be non-ECs. This discrepancy should be addressed.

2. In Figure 5B, GFP expression increases in ECs from WT mice after doxycycline (DOX) removal. The authors should clarify whether GFP is a reliable proxy for progerin expression in Prog-Tg cells and explain this unexpected result.

Referee #3 Review

Report for Author:

In this paper, the authors test the hypothesis that the Progerin mutation of LMNA causes ER stress in endothelial cells (ECs) and this ER stress might contribute to the phenotype that this group previously observed in the transgenic animals from which the mice are derived. Using characterization of a few selected UPR markers, the authors come to the conclusions that (1) Progerin does not cause ER stress in ECs except for only very initially after its expression, (2) that Progerin-expressing ECs remain competent for UPR signaling; and (3) that signs of UPR activation evident in the hearts of transgenic G609G/+ animals cannot be explained by a contribution from the ECs. Given that there is evidence for UPR activation in the latter, it is of interest to the Progerin community to know whether ECs contribute to this signal.

Of the 3 conclusions listed above, #2 seems to be fairly straightforwardly true. #1 and #3 are more problematic. I think

both conclusions are probably correct, especially since Progerin is not expressed in the ER, so even in cell types in which it does apparently cause ER stress, the effect is likely to be highly indirect. But the central problem with the first and third conclusions is that definitively demonstrating a negative—that is, lack of a contribution of ER stress—requires a much more rigorous examination than is provided here, particularly given lack of direct genetic tests of a UPR contribution (for example, whether EC-specific ablation of IRE1, PERK, or ATF6 affects the phenotype—though note that I am not suggesting that the authors conduct those experiments here).

There are also some issues including that the authors' assumptions about how the UPR works and how to interpret some of their readouts are dated and/or incorrect.

Major conceptual points

- I could imagine a scenario in which Progerin expression elicits a steady but slight activation of the UPR that, over time, leads to rather consequential cellular dysregulation. It is worth noting that the EC experiments here use cells from very young animals (8-21d). In fact, it is not clear that animals that age have any appreciable Progerin-dependent phenotype, so in that sense it is perhaps not surprising that the cells show no obvious UPR activation from the few markers analyzed. Other than the fact that they do appear to express Progerin, it isn't clear that the ECs show in any way negative effects of Progerin expression. One might expect that evidence for UPR activation might be much more robust in ECs of older animals, which as far as I can tell was not tested in either the Prog-Tg or the G609G/+ animals
- The apparent lack of a UPR from bulk RNA analysis of lung, heart (fig. EV1), and aorta (Fig. 6C) is difficult to interpret. Presumably ECs make up a minor population of those organs (~10% at least in the heart from what I gather), so it isn't clear that a UPR signal, if it existed in that relatively minor population, would be strong enough to be detected above the baseline, especially with an n=3.
- Similarly to the point above, it is fairly clear that there is a detectable UPR (or ISR; more on this below) signal in the Aortae of G609G/+ animals (Fig. 6B). However, again due to the issue of relative cell scarcity, the data in 6C don't allow one to conclude that all UPR signaling in the G609G/+ animals emanates from non-ECs.
- mRNAs tend to be the most sensitive readout for UPR activation, but it is also true that some of the proteins regulated by the UPR are much more stable and might be more prone to dysregulation. I could easily imagine that chaperones in particular like HSPA5, HSP90B1, or CALR might be altered by Progerin expression even when changes in their mRNAs aren't evident.
- Ultimately, if the authors want to be able to more definitively exclude that UPR signaling occurs in ECs in response to Progerin expression, they would need to examine ECs at different ages, and also in vivo, examining in particular for the latter relative expression of ER chaperones and other UPR markers and do so in an extremely systematic way that allows what might be minor differences to be reliably detected and quantified.

I don't raise these points because I think the authors are wrong—as I said above, I think their conclusion is probably correct, but the present manuscript does not meet the needed burden of proof.

Other substantial points:

- Hsp90b1, Hspa5, and Edem1 are not particularly diagnostic for IRE1 activation. See PMID:27984733 for a more recent and more definitive categorization of which genes are better diagnostic markers for each UPR pathway. Beyond Xbp1 splicing, translocon components like Sec61a1, Sec61b, and Ssr1 among others genes read out well on IRE1 activation. In addition, the canonical RIDD target Bloc1s1 should be assessed.
- One possibility (perhaps more plausible than a UPR) is that Progerin induces a more general ISR, maybe through PERK or maybe through one of the other ISR kinases. A more exhaustive analysis of ISR genes (again, the paper cited above is a good resource) as well as characterization of the phosphorylation state of eIF2a would be needed.
- Fig. 2F, expression of Atf6 mRNA says nothing about the activation status of the ATF6 pathway.
- Fig. 2G is misinterpreted. The band that the authors are observing in the 4th lane is not ATF6(N), which is much smaller (~55 kDa). Rather, it is the unglycosylated form of ATF6 that is observed upon inhibition of glycosylation with TM. This band says nothing about the activation status of ATF6. In fact, given that endogenous mouse ATF6 is notoriously difficult to detect with convincing specificity, I'd be surprised if the authors see ATF6(N) even in their TM control.
- Figs. 2C and 3C—the dynamic range of these flow assays is very small. Even in the presence of 2.5 ug/ml TM—an extremely robust stressor that is roughly 100-1000-fold higher than needed to activate the UPR in most cell types, the increase in XBP1 and ATF4 fluorescence is very modest, less than 10-fold. If one assumes that such a stress is much, much more potent than whatever stress Progerin might elicit, then it becomes very difficult to interpret the negative result. And, moreover, it even appears as if Prog-Tg does indeed result in a rightward shift for ATF4 fluorescence—

though no statistical analysis is provided, this is one of the bits of data that might favor ISR activation even if not UPR activation.

- In general the manuscript is not clear on what the n-numbers are and what level of replication they represent (for example Fig. 2A, not clear what each dot represents)
- Sex of animals used is not as far as I can tell specified.

Minor points:

- Why does the LMN1 antibody only detect Progerin? Is it actually specific for the mutant form, or is Progerin so highly overexpressed in the transgene-expressing animal that the relative signal in wild-type animals is undetectable?
- Statistical significance stars should be enlarged so they aren't confused with data points.
- The images would be more convincing if more than one single cell were shown-it isn't clear how unbiased the authors are being in their image selection.
- The following statement doesn't make sense: "Only Gadd34 was upregulated in Prog-Tg versus WT ECs (Fig 3B), but Gadd34 is also involved in other integrated stress response pathways, namely senescence" I say this because all stresses that converge on the integrated stress response lead to eIF2a phosphorylation and upregulation of ISR-dependent genes including Gadd34, Chop, etc.
- Authors claim "Furthermore, Tunicamycin induced XBP1s translocation" This is not the correct conclusion. The data don't suggest that XBP1 localization changes from cytoplasmic to nuclear. Rather, the amount of nuclear XBP1 increases, which is most consistent with an increase in expression of XBP1 protein (and with the accepted understanding of how the IRE1/XBP1 pathway works)
- The discussion could be condensed substantially, as in its current form it expounds greatly on fairly fine threads of evidence.

August 19, 2025

Re: Life Science Alliance manuscript #LSA-2025-03485-T

Dr. Roland Foisner
Medical University of Vienna
Max Perutz Labs
Dr. Bohr-Gasse 9
Vienna 1030
Austria

Dear Dr. Foisner,

Thank you for transferring your manuscript entitled "The unfolded protein response in progeria arteries originates from non-endothelial cell types" to Life Science Alliance. As indicated in the decision letter from another journal, we invite you to submit a revised manuscript with these changes:

- Temper the main conclusion, in light of the fact that the tissue used in these analyses was from young, unstressed mice (Reviewer 3).
- Clarify that results here focused on lung ECs which may differ from vascular/aortic ECs.
- Address Reviewer 2 point 3 on reporting the expression levels of LmnaG609G vs Progerin in the two mouse models, or clearly state that differences in expression were not evaluated.

When submitting the revision, please include a letter addressing the points noted above. While you are revising your manuscript, please also attend to the below editorial points to help expedite the publication of your manuscript. Please direct any editorial questions to the journal office.

Thank you for this interesting contribution to Life Science Alliance. We are looking forward to receiving your revised manuscript.

Sincerely,

- A letter addressing the reviewers' comments point by point.
- An editable version of the final text (.DOC or .DOCX) is needed for copyediting (no PDFs).
- High-resolution figure, supplementary figure and video files uploaded as individual files: See our detailed guidelines for preparing your production-ready images, <https://www.life-science-alliance.org/authors>
- Summary blurb (enter in submission system): A short text summarizing in a single sentence the study (max. 200 characters including spaces). This text is used in conjunction with the titles of papers, hence should be informative and complementary to the title and running title. It should describe the context and significance of the findings for a general readership; it should be written in the present tense and refer to the work in the third person. Author names should not be mentioned.
- By submitting a revision, you attest that you are aware of our payment policies found here: <https://www.life-science-alliance.org/copyright-license-fee>

B. MANUSCRIPT ORGANIZATION AND FORMATTING:

Point-by-point response to comments of editor and reviewers

Editor of LSA:

As indicated in the decision letter from another journal, we invite you to submit a revised manuscript with these changes:

- Temper the main conclusion, in light of the fact that the tissue used in these analyses was from young, unstressed mice (Reviewer 3).

RESPONSE:

We agree that most of our *in vitro* analyses were done with primary lung endothelial cells, isolated from young unstressed mice and expanded in culture. We could not perform these *in vitro* experiments with cells from old mice, as one cannot expand old endothelial cells in culture.

In order to respond to the request of the editor, we have included **new data** from older tissues in **new Fig 6**, showing that endothelial cells enriched from heart tissue isolated from older mice (6 months) still display no UPR with trends to even lower values. Additionally, aorta samples from *Prog-Tg* and *Lmna*^{G609G/+} mice, shown in **Fig 6C** and **Fig 7B**, are derived from older mice (2-3 months), the age used in previous reports showing elevated UPR signaling in aortas from HGPS mice (2-4 month-old mice in (Hamczyk *et al*, 2019); and 3-5 month-old ones in (Vidak *et al*, 2023)). It has to be noted that 50% of *Prog-Tg* animals die at 6 months (Osmanagic-Myers *et al*, 2019) and *Lmna*^{G609G/+} mice die at around 7 months (Osorio *et al*, 2011).

In addition to the new data, we also clearly indicate in the text that the *in vitro* data were derived from young endothelial cells (page 7, and in a new paragraph on page 12), and we tempered the conclusions accordingly.

- Clarify that results here focused on lung ECs which may differ from vascular/aortic ECs.

RESPONSE:

We now clarify in the text that all our *in vitro* cell culture experiments were done with lung ECs (page 7 and new paragraph on page 12), and additionally, we added **new data** in **new Fig 6** showing ECs derived from heart of old mice and whole heart and aortic tissue lysates from old mice.

In addition, **new analyses** of publicly available scRNA-Seq data sets of aortic cell populations from *Lmna*^{+/+} and *Lmna*^{G609G/G609G} mice (Barettino *et al*, 2024) revealed no upregulation of ER stress- and UPR-related genes in EC populations, while VSMCs show activation of UPR genes. These **new analyses** are shown in **new Fig 8**. Thus, the revised version of our manuscript shows no UPR upregulation in 3 different EC populations (lung, heart, aorta).

- Address Reviewer 2 point 3 on reporting the expression levels of *Lmna*^{G609G} vs Progerin in the two mouse models, or clearly state that differences in expression were not evaluated.

RESPONSE:

We have performed Western blotting of cell lysates of lung ECs derived from *Prog-Tg*, *Lmna*^{G609G/+} and wildtype cells using an anti-lamin A/C antibody, revealing the levels of progerin in ECs in these two different mouse models in comparison to progerin levels in human HGPS fibroblasts. We now show these Western blot results in the **new Fig 7D**.

Point-by-point response to reviewers

Referee #1:

This is a manuscript by Silva et al exploring the molecular pathogenesis of Hutchinson-Gilford progeria syndrome (HGPS), which is a rare, premature aging disorder. One of the hallmarks of HGPS is accelerated cardiovascular disease, in a form of severe atherosclerosis that underlies arterial stiffening as well as myocardial infarction if left untreated. As prior studies have linked dysregulated activation of unfolded protein response (UPR) in vascular dysfunction, the authors focused specifically on whether progerin expression in endothelial cells (ECs) can elicit ER stress directly. Using progerin-expressing ECs from an EC specific progeria mouse model (Prog-Tg), the authors investigated the three canonical UPR pathways. They found that constitutive progerin expression in ECs alone does not activate UPR despite retaining the ability to do so. Furthermore, they show that arteries from mice ubiquitously expressing progerin display UPR activation but not in EC-specific expression, demonstrating that UPR in HGPS originates from non-endothelial arterial cell populations.

The strength of the manuscript is that the authors rigorously examined previously identified disease mechanisms in a cell type specific-manner in the various cells that form the vasculature. This has led to a deeper understanding of how vascular complications arise in HGPS - that dysregulated UPR is not directly linked to progerin expression in ECs and likely occurs other cell types of the vasculature. A major weakness is that it does not push the boundaries of our current understanding. No novel pathways or molecular pathogenesis has been linked to HGPS. Despite this, the study findings are important, timely, and provide sufficient conceptual advance critical for understanding the pathogenesis of cardiovascular complications in HGPS.

RESPONSE:

We thank the reviewer for the positive comments and conclusions. We agree that we do not show any new disease pathway, but we feel it is important for the progeria research community to dissect cell-type differences regarding UPR. This information is also highly important in view of any potential UPR-based therapies to emphasize that targeting UPR will presumably not ameliorate EC dysfunction in the disease context.

Points to consider:

1) It's not clear to be what's the purpose of presenting data in Fig, EV1C and EV1D. The text descriptor states "it is important to note that these samples had lower progerin expression than the cultured ECs, given that non-ECs in these tissues do not express progerin" but the data shows neither of these.

RESPONSE:

Thank you for bringing up this point. Initially, our purpose was to show that progerin transcript levels are lower in tissue samples compared to isolated endothelial cells by comparing data shown in Fig. 1D and Fig. EV1. However, we agree with the reviewer's criticisms and have removed these data from the manuscript.

2) Why use ECs from the lungs? Wouldn't it be more appropriate to isolate ECs from myocardial vasculature? Some explanation would be helpful.

RESPONSE:

We understand the reviewer's concerns. ECs isolated from lung propagate in culture more efficiently, rendering higher cell yield compared to those isolated from cardiac tissue. We have previously shown that ECs from lung and heart behave very similarly, with both upregulating senescence pathways linked to a pro-inflammatory and pro-fibrotic senescence-associated secretory phenotype (Manakanatas *et al*, 2022).

However, in response to the concern of the reviewer, we now also present data from EC populations derived from heart samples (**new Fig 6**). In addition, we also analyzed publicly available single-cell RNA-Seq data sets, revealing no upregulation of the UPR response in aortic ECs, unlike in VSMCs (**new Fig 8**). Thus, we demonstrate a lack of the UPR in 3 different EC populations (lung, heart, aorta).

Please see also our comment to point 2 of the editor.

3) It's a missed opportunity that the lung wasn't stressed in any way to ascertain whether any sort of physiological stress can activate UPR in progerin expressing ECs.

RESPONSE:

We agree that the application of physiological stress to the lung would be an interesting approach. However, this would include additional animal experimentation that goes beyond the scope of this paper, which aims at reporting the lack of UPR in progerin-expressing ECs under basic, unstressed conditions. It should be mentioned though that we have included data from older animals in **new Fig 6** of the revised manuscript. Additionally, we stressed cultured ECs with Tunicamycin, showing a similar response of *Prog-Tg* and *WT* ECs (**Fig 4**).

4) Acute progerin expression leading to XBP1 splicing is interesting. Although the use of cells from in vivo model to demonstrate this effect is commendable, it is unclear whether the nuclear enrichment of XBP1s is due to progerin expression itself or due to stress inherent to any transgene overexpression.

RESPONSE:

We agree that our results do not completely rule out that XBP1 splicing is activated upon expressing any transgene, rather than specifically by progerin transgene expression. Yet, UPR is highly specific to folding defects in the ER, and therefore, we do not expect that any transgene that is in the nucleus would activate the UPR signaling. Moreover, our previous studies have shown that endothelial expression of a wildtype lamin A transgene (*LA-Tg*) does not lead to any of the phenotypes observed in *Prog-Tg* mice, such as cardiac hypertrophy, diastolic dysfunction, interstitial cardiac and perivascular fibrosis (Osmanagic-Myers *et al.*, 2019). Furthermore, lamin A-overexpressing ECs do not activate the senescence and SASP pathways found in progerin-expressing ECs (Manakanatas *et al.*, 2022). These data indicate that lamin A overexpression does not impair cellular physiology. Thus, we did not further analyze UPR pathways upon lamin A overexpression.

5) The predominant localization of GFP in ECs from Pro-Tg mice is somewhat surprising. Typically, GFP without any localization signal (expressed via IRES) would look like diffuse GFP staining the whole cell. However, authors see distinct nuclear rim staining, reminiscent of lamin A-GFP fusion protein. Some explanation would be helpful.

RESPONSE:

We do not show any GFP staining in immunofluorescence images throughout the manuscript. All microscopic images showing peripheral nuclear staining were generated using an anti-human lamin A antibody, which detects the human progerin transgene. GFP, which is expressed from an IRES in the transgene construct, was used for FACS analyses only, as GFP intensities can be used as a proxy for progerin expression levels.

6) Although I appreciate the effort to perform kinetic analysis (Fig. EV2) to determine the optimal timepoint for maximal UPR gene expression (Fig. 4A-C), the approach taken may miss potentially altered expression kinetics caused by progerin.

RESPONSE:

To respond to the reviewer, we have now added the kinetics of progerin-expressing ECs in the **updated Fig S1B**, which show a very similar pattern of UPR activation kinetics compared to that of *WT* ECs.

Referee #2:

Silva and colleagues investigated the effects of progerin in endothelial cells (ECs) on the unfolded protein response (UPR) in the context of Hutchinson-Gilford progeria syndrome (HGPS), a premature aging disorder characterized by accelerated cardiovascular disease (CVD). While previous studies have implicated UPR activation in vascular smooth muscle cells (VSMCs) and fibroblasts, the contribution of ECs to this process remains unclear.

The authors used a transgenic mouse model expressing progerin specifically in ECs (Prog-Tg) and assessed UPR activation through canonical ER stress sensors (IRE1 α , PERK, and ATF6). They found that sustained progerin expression in ECs does not induce a robust UPR at baseline, although the cells remain responsive to tunicamycin-induced ER stress. Interestingly, acute progerin expression does activate UPR through the IRE1 α pathway, suggesting a potential adaptive mechanism that resets UPR signaling to basal levels under chronic progerin exposure. The investigations were extended to *in vivo* models. They confirmed earlier studies showing UPR activation in aortas of *Lmna*^{G609G} mice (that express progerin in all cells) but they did not observe UPR activation in Prog-Tg mice, supporting the idea that UPR activation in HGPS arteries originates from non-endothelial cells.

The manuscript is clearly written, and the data are well presented. However, several points should be addressed to strengthen the conclusions.

RESPONSE:

We appreciate the positive evaluation of the manuscript.

Comments:

1. The authors should clarify the rationale for using lung ECs rather than ECs from progeria-affected arteries (e.g., aorta). The strategy to isolate ECs (ICAM-2 expression) likely yields predominantly capillary ECs. Are these representative of arterial ECs in terms of gene expression, function, or disease relevance in HGPS? The authors should discuss whether their findings can be generalized to arterial ECs.

RESPONSE:

We used lung ECs because the isolation from this tissue is much more efficient, with much higher yields compared to isolation from the heart or aorta. Also, ECs from cardiac tissue are hard to propagate *in vitro* in culture. Moreover, previous experiments have shown that ECs derived from the lung behave similarly to those from the heart in progeria mice (Manakanatas *et al.*, 2022).

Thus, based on the limited number of progerin-expressing tissues available, we performed all *in vitro* experiments with lung ECs derived from young mice. However, in response to the reviewer, we have also added **new data** in **new Fig 6**, showing EC populations enriched from heart tissue. In addition, we also analyzed publicly available single-cell RNA-Seq data sets, revealing no upregulation of the UPR response in aortic ECs, unlike in VSMCs (**new Fig 8**). Thus, we demonstrate a lack of the UPR in 3 different EC populations (lung, heart, aorta).

Please see also our response to comment 2 of reviewer 1.

2. ICAM-2 is expressed by multiple EC subtypes (capillary, arterial, venous, and possibly lymphatic). The composition of the lung EC preparation should be better characterized. The

authors should clarify the relative proportions of these subtypes and whether progerin expression alters this composition—for example, by reducing the proportion of arterial ECs.

RESPONSE:

We agree that ICAM-2 is expressed across multiple endothelial subtypes, including arterial, venous, and capillary ECs. However, we would like to note that endothelial cells in culture, even when initially isolated from specific vascular beds, tend to lose many of their *in vivo*-specific transcriptional and functional characteristics, resulting in a more homogeneous endothelial phenotype (Lacorre *et al.*, 2004). Consequently, while we can confirm that our lung EC preparation expresses pan-endothelial markers (e.g. CD31, VE-cadherin, ICAM-2), it is unlikely that distinct vascular subtypes are faithfully maintained under standard culture conditions.

Nevertheless, we agree that a detailed analysis of the EC subtypes would be interesting, but these analyses would require a lot of additional experiments and animals going beyond the scope of the manuscript.

However, in the revised version of the manuscript, we provide additional **new data** from heart and aortic ECs (**new Figs 6-8**) and thus show three types of EC populations (lung, heart, aorta) in this study.

3. The conclusion that UPR activation is cell type-specific assumes comparable progerin expression in ECs across the $Lmna^{G609G}$ and Prog-Tg models. This assumption should be experimentally validated. Ideally, progerin levels should be measured in aortic ECs. If this is not feasible, Western blotting of lung ECs using an antibody that cross-reacts with both human and mouse progerin should be performed.

RESPONSE:

Unfortunately, we can't do Western blots from aortic EC due to low amounts of ECs in the tissue that would require too many animals, which would not be in accordance with 3R principles. To address this concern of the reviewer, we show Western blots of cell lysates of lung ECs derived from Prog-Tg, $Lmna^{G609G/+}$ and wildtype cells using an anti-lamin A/C antibody (**new Fig 7D**). These data reveal the levels of progerin in ECs in the two different mouse model in comparison to progerin levels in human HGPS fibroblasts.

Please see also the response to comment 3 of the editor.

4. The hypothesis that constitutive progerin expression induces an adaptive mechanism to reduce baseline UPR activation is novel and intriguing. To validate this idea, control studies with ECs isolated from control LA-Tg mice (that express GFP and wild type lamin A) should be included. The authors have used this model in previous studies (Osmanagic-Myers *et al.*, 2019).

RESPONSE:

Indeed, we have previously used the LA-Tg mice as a control, but did not find any of the phenotypes associated with the Prog-Tg mice (Manakanatas *et al.*, 2022; Osmanagic-Myers *et al.*, 2019). Therefore, we stopped breeding these animals in our mouse facilities due to space limitations. Re-establishing this mouse line would take a lot of time and significantly delay the revision process, most likely without providing new insights.

Minor Comments:

1. The claim that primary lung ECs are highly pure is not fully supported by Figure 1F, where approximately 45% of cells in the image (5/11) appear to be non-ECs. This discrepancy should be addressed.

RESPONSE:

We agree that **Fig 1F** also shows some non-endothelial cells in this particular field. However, all other analyses (qPCR and FACs), which take into account the entire cell population, support our conclusion. In order to respond to the reviewer, we have exchanged **Fig 1F** with a new microscopic

image with a larger representative view that better reflects the results of the quantitative assays. We have also exchanged **Fig 1G** to match the field size of the **new Fig 1F**.

2. In Figure 5B, GFP expression increases in ECs from WT mice after doxycycline (DOX) removal. The authors should clarify whether GFP is a reliable proxy for progerin expression in *Prog-Tg* cells and explain this unexpected result.

RESPONSE:

The reviewer is right in saying that there is a subtle shift in the FACS GFP signal during Doxycycline treatment. However, this is likely due to an increase in the cells' green autofluorescence, as this slight shift is seen for all samples, including *WT*, which do not express GFP. Autofluorescence can vary with cell metabolic state and culture conditions, which may explain the minor increase following doxycycline withdrawal. Importantly, the GFP signal observed in *Prog-Tg* cells after doxycycline removal can be clearly detected as a separate peak. Thus, the specific upregulation of GFP as a proxy for progerin expression can clearly be distinguished from an unspecific, subtle increase in green signal during incubation and prolonged cell culture. As a response to the reviewer's point, we added this explanation in the text (page 11).

To further respond to the reviewer's concern, we show below for the reviewer a comparison of the FACS GFP signal with a progerin immunofluorescence staining of the same *Prog-Tg* sample and the same *WT* control, supporting our conclusion that GFP is a good proxy for progerin expression

[Figure removed by editorial staff per authors' request].

Referee #3:

In this paper, the authors test the hypothesis that the Progerin mutation of LMNA causes ER stress in endothelial cells (ECs) and this ER stress might contribute to the phenotype that this group previously observed in the transgenic animals from which the mice are derived. Using characterization of a few selected UPR markers, the authors come to the conclusions that (1) Progerin does not cause ER stress in ECs except for only very initially after its expression, (2) that Progerin-expressing ECs remain competent for UPR signaling; and (3) that signs of UPR activation evident in the hearts of transgenic G609G/+ animals cannot be explained by a contribution from the ECs. Given that there is evidence for UPR activation in the latter, it is of interest to the Progerin community to know whether ECs contribute to this signal.

RESPONSE:

We thank the reviewer for appreciating the importance of our analyses for the progerin community.

Of the 3 conclusions listed above, #2 seems to be fairly straightforwardly true. #1 and #3 are more problematic. I think both conclusions are probably correct, especially since Progerin is not expressed in the ER, so even in cell types in which it does apparently cause ER stress, the effect is likely to be highly indirect. But the central problem with the first and third conclusions is that definitively demonstrating a negative—that is, lack of a contribution of ER stress—requires a much more rigorous examination than is provided here, particularly given lack of direct genetic tests of a UPR contribution (for example, whether EC-specific ablation of IRE1, PERK, or ATF6 affects the phenotype—though note that I am not suggesting that the authors conduct those experiments here).

RESPONSE:

We thank the reviewer for her/his overall evaluation. We agree that a more rigorous validation of our conclusion would be beneficial, such as genetically targeting UPR components, but we also feel that this would require a lot of additional experiments that go beyond the scope of the manuscript.

There are also some issues including that the authors' assumptions about how the UPR works and how to interpret some of their readouts are dated and/or incorrect.

Major conceptual points

- I could imagine a scenario in which Progerin expression elicits a steady but slight activation of the UPR that, over time, leads to rather consequential cellular dysregulation. It is worth noting that the EC experiments here use cells from very young animals (8-21d). In fact, it is not clear that animals that that age have any appreciable Progerin-dependent phenotype, so in that sense it is perhaps not surprising that the cells show no obvious UPR activation from the few markers analyzed. Other than the fact that they do appear to express Progerin, it isn't clear that the ECs show in any way negative effects of Progerin expression. One might expect that evidence for UPR activation might be much more robust in ECs of older animals, which as far as I can tell was not tested in either the Prog-Tg or the G609G/+ animals

RESPONSE:

Thank you for the insightful remarks. Please note that ECs isolated from very young *Prog-Tg* animals already develop senescence, senescence-associated proinflammatory effects, as well as profibrotic effects in co-culture, as shown by our studies (Fleischhacker *et al*, 2024; Manakanatas *et al.*, 2022). Thus, the analyses were done in ECs isolated from young animals since one cannot isolate enough ECs and obtain sufficient propagation in culture from old mice necessary for *in vitro* experiments. However, all tissues analysed (heart and aorta) were derived from older mice. Additionally, we have now added **new data** in **new Fig 6** showing analyses of ECs derived from heart of old mice.

Please see also our response to comment 1 of the editor.

- The apparent lack of a UPR from bulk RNA analysis of lung, heart (fig. EV1), and aorta (Fig. 6C) is difficult to interpret. Presumably ECs make up a minor population of those organs (~10% at least in the heart from what I gather), so it isn't clear that a UPR signal, if it existed in that relatively minor population, would be strong enough to be detected above the baseline, especially with an n=3.

RESPONSE:

We agree with the reviewer and have therefore used isolated ECs rather than whole tissues for our analyses, facing the caveat that ECs cannot be isolated from old mice and expanded in cell culture. To overcome these problems, we have performed additional experiments, sorting ECs from heart of older mice (6 months) and performing the analysis without previous expansion in cell culture. We have added this analysis in **new Fig 6**. Unfortunately, this analysis cannot be done in the aorta due to the limited number of ECs obtained in the isolation that would require many animals and would not be in accordance with 3R principles.

However, we have analyzed publicly available single-cell RNA sequencing data sets of aortic cell types, revealing no upregulation of UPR in aortic ECs in contrast to obvious effects in VSMCs (**new Fig 8**).

Please see also our response to comments 1 and 2 of the editor.

- Similarly to the point above, it is fairly clear that there is a detectable UPR (or ISR; more on this below) signal in the Aortae of G609G/+ animals (Fig. 6B). However, again due to the issue of relative cell scarcity, the data in 6C don't allow one to conclude that all UPR signaling in the G609G/+ animals emanates from non-ECs.

RESPONSE:

We agree with the reviewer's concern, but unfortunately, the approach described above to isolate ECs and non-ECs from the aorta cannot be done due to the limited number of cells.

- mRNAs tend to be the most sensitive readout for UPR activation, but it is also true that some of the proteins regulated by the UPR are much more stable and might be more prone to dysregulation. I could easily imagine that chaperones in particular like HSPA5, HSP90B1, or CALR might be altered by Progerin expression even when changes in their mRNAs aren't evident.

RESPONSE:

We have performed immunofluorescence staining and quantitative Western blotting for HSPA5 (BiP), but could not observe any difference in protein levels in *Prog-Tg* vs *WT* ECs. We show the blot here for the reviewer (**Reviewer Fig 2**).

Reviewer Figure 2 – BiP protein levels are unchanged in ECs from *Prog-Tg* mice.

(A) Immunoblot analysis of *WT* and *Prog-Tg* EC lysates stained with an anti-BiP antibody (upper) and γ -Tubulin antibody (lower). Cells were treated with Tunicamycin (2.5 μ g/ml, 8h) and the respective DMSO control. (B) Quantification of BiP/ γ -Tubulin protein band intensities in *WT* and *Prog-Tg* ECs (DMSO), based on the immunoblot shown in (A). Intensity levels were normalized to *WT* sample.

- Ultimately, if the authors want to be able to more definitively exclude that UPR signaling occurs in ECs in response to Progerin expression, they would need to examine ECs at different ages, and also in vivo, examining in particular for the latter relative expression of ER chaperones and other UPR markers and do so in an extremely systematic way that allows what might be minor differences to be reliably detected and quantified.

RESPONSE:

In addition to data shown for ECs isolated from young mice, we included now in **new Fig 6A** of the revised manuscript data on ECs derived from hearts of older mice with analysis of chaperones and all key UPR markers.

I don't raise these points because I think the authors are wrong-as I said above, I think their conclusion is probably correct, but the present manuscript does not meet the needed burden of proof.

Other substantial points:

- Hsp90b1, Hspa5, and Edem1 are not particularly diagnostic for IRE1 activation. See PMID:27984733 for a more recent and more definitive categorization of which genes are better diagnostic markers for each UPR pathway. Beyond Xbp1 splicing, translocon components like Sec61a1, Sec61b, and Ssr1 among other genes read out well on IRE1 activation. In addition, the canonical RIDD target Bloc1s1 should be assessed.

RESPONSE:

We want to clarify that in this study, we do not aim to identify specific activation of each UPR branch. Our goal was to determine whether there is induction of the UPR in ECs expressing progerin, which was already reported for other cell types (Hamczyk *et al.*, 2019; Vidak *et al.*, 2023). We do not see increased levels of *Xbp1s*, *Edem1*, and *Hspa5* mRNAs (**Fig 2**) and selected UPR marker proteins (**Fig 2** and **Fig 3**) upon progerin expression in ECs, indicating that there is not a robust UPR activation in our experimental system.

- One possibility (perhaps more plausible than a UPR) is that Progerin induces a more general ISR, maybe through PERK or maybe through one of the other ISR kinases. A more exhaustive

analysis of ISR genes (again, the paper cited above is a good resource) as well as characterization of the phosphorylation state of eIF2a would be needed.

RESPONSE:

ISR induces robust translation of *Atf4* mRNA, which encodes for a transcription factor that regulates the expression of downstream targets such as *Chop* and *Gadd34*. In any of the RT-qPCR analyses, we do not observe an increase in the transcripts, indicating that the PERK branch is not activated in those cells. We did not see any indication of activation of the PERK kinase. Testing other ISR markers would go beyond the scope of this manuscript.

- Fig. 2F, expression of *Atf6* mRNA says nothing about the activation status of the ATF6 pathway.

RESPONSE:

We agree with the reviewer, but we wanted to show a full analysis, particularly since *Atf6* transcript levels were also shown in other reports claiming activation of the UPR in progeria (Vidak *et al.*, 2023).

- Fig. 2G is misinterpreted. The band that the authors are observing in the 4th lane is not ATF6(N), which is much smaller (~55 kDa). Rather, it is the unglycosylated form of ATF6 that is observed upon inhibition of glycosylation with TM. This band says nothing about the activation status of ATF6. In fact, given that endogenous mouse ATF6 is notoriously difficult to detect with convincing specificity, I'd be surprised if the authors see ATF6(N) even in their TM control.

RESPONSE:

We thank the reviewer for bringing this up. We checked for a band at around 55 kDa in our blot, as indicated by the reviewer, but could not detect any new band upon tunicamycin treatment. Thus, we removed the blot and all the accompanying text in the manuscript.

- Figs. 2C and 3C-the dynamic range of these flow assays is very small. Even in the presence of 2.5 ug/ml TM-an extremely robust stressor that is roughly 100-1000-fold higher than needed to activate the UPR in most cell types, the increase in XBP1 and ATF4 fluorescence is very modest, less than 10-fold. If one assumes that such a stress is much, much more potent than whatever stress Progerin might elicit, then it becomes very difficult to interpret the negative result. And, moreover, it even appears as if Prog-Tg does indeed result in a rightward shift for ATF4 fluorescence-though no statistical analysis is provided, this is one of the bits of data that might favor ISR activation even if not UPR activation.

RESPONSE:

We agree with the reviewer that the response to progerin expression, if any, may be extremely subtle and undetectable by the assays. However, the main point of the manuscript was to show that a robust activation of the UPR, as reported for VSMCs (Hamczyk *et al.*, 2019), is not detectable in progerin-expressing ECs.

- In general the manuscript is not clear on what the n-numbers are and what level of replication they represent (for example Fig. 2A, not clear what each dot represents)

RESPONSE:

We changed the figure legends to specify what n represents for each case (number of biological replicates, number of cells, number of mice).

- Sex of animals used is not as far as I can tell specified.

RESPONSE:

We have now added a statement indicating that we did not consider the sex of animals (page 20). This was not feasible due to the low number of mice available for this study.

Minor points:

- Why does the LMN1 antibody only detect Progerin? Is it actually specific for the mutant form, or is Progerin so highly overexpressed in the transgene-expressing animal that the relative signal in wild-type animals is undetectable?

RESPONSE:

Prog-Tg mice express a human progerin transgene, which is detectable with an anti-human lamin A antibody that does not detect mouse lamin A (Manakanatas *et al.*, 2022).

- Statistical significance stars should be enlarged so they aren't confused with data points.

RESPONSE:

We have enlarged the statistical significance stars in all Figures.

- The images would be more convincing if more than one single cell were shown-it isn't clear how unbiased the authors are being in their image selection.

RESPONSE:

The reviewer is correct that the microscopic images may be somewhat biased, but in order to avoid this for our conclusions, we always show the quantification of data next to the image, which represents an unbiased quantitative analysis.

- The following statement doesn't make sense: "Only Gadd34 was upregulated in Prog-Tg versus WT ECs (Fig 3B), but Gadd34 is also involved in other integrated stress response pathways, namely senescence" I say this because all stresses that converge on the integrated stress response lead to eIF2a phosphorylation and upregulation of ISR-dependent genes including Gadd34, Chop, etc.

RESPONSE:

ISR is composed of 4 kinases that are activated upon different stress inputs. PERK, PRK, GCN2, HRI. They all sense different stresses and phosphorylate eIF2alpha. We do not completely understand the point of the reviewer. GADD34 is regulated by CHOP and we do not see CHOP induction, but it may be regulated by some other means here (Marciniak *et al.*, 2004).

- Authors claim "Furthermore, Tunicamycin induced XBP1s translocation" This is not the correct conclusion. The data don't suggest that XBP1 localization changes from cytoplasmic to nuclear. Rather, the amount of nuclear XBP1 increases, which is most consistent with an increase in expression of XBP1 protein (and with the accepted understanding of how the IRE1/XBP1 pathway works)

RESPONSE:

We have rephrased this sentence according to the suggestion of the reviewer (page 10).

- The discussion could be condensed substantially, as in its current form it expounds greatly on fairly fine threads of evidence.

RESPONSE:

We have tried to streamline the discussion according to the suggestion of the reviewer.

References:

Barettino A, Gonzalez-Gomez C, Gonzalo P, Andres-Manzano MJ, Guerrero CR, Espinosa FM, Carmona RM, Blanco Y, Dorado B, Torroja C *et al* (2024) Endothelial YAP/TAZ activation promotes atherosclerosis in a mouse model of Hutchinson-Gilford progeria syndrome. *J Clin Invest* 134

Fleischhacker V, Milosic F, Bricelj M, Kuhrer K, Wahl-Figlash K, Heimel P, Diendorfer A, Nardini E, Fischer I, Stangl H *et al* (2024) Aged-vascular niche hinders osteogenesis of mesenchymal stem cells through paracrine repression of Wnt-axis. *Aging Cell* 23: e14139

Hamczyk MR, Villa-Bellosta R, Quesada V, Gonzalo P, Vidak S, Nevado RM, Andres-Manzano MJ, Misteli T, Lopez-Otin C, Andres V (2019) Progerin accelerates atherosclerosis by inducing endoplasmic reticulum stress in vascular smooth muscle cells. *EMBO Mol Med* 11

Lacorre DA, Baekkevold ES, Garrido I, Brandtzaeg P, Haraldsen G, Amalric F, Girard JP (2004) Plasticity of endothelial cells: rapid dedifferentiation of freshly isolated high endothelial venule endothelial cells outside the lymphoid tissue microenvironment. *Blood* 103: 4164–4172

Manakanatas C, Ghadge SK, Agic A, Sarigol F, Fichtinger P, Fischer I, Foisner R, Osmanagic-Myers S (2022) Endothelial and systemic upregulation of miR-34a-5p fine-tunes senescence in progeria. *Aging (Albany NY)* 14: 195–224

Marciniak SJ, Yun CY, Oyadomari S, Novoa I, Zhang Y, Jungreis R, Nagata K, Harding HP, Ron D (2004) CHOP induces death by promoting protein synthesis and oxidation in the stressed endoplasmic reticulum. *Genes Dev* 18: 3066–3077

Osmanagic-Myers S, Kiss A, Manakanatas C, Hamza O, Sedlmayer F, Szabo PL, Fischer I, Fichtinger P, Podesser BK, Eriksson M *et al* (2019) Endothelial progerin expression causes cardiovascular pathology through an impaired mechanoresponse. *J Clin Invest* 129: 531–545

Osorio FG, Navarro CL, Cadinanos J, Lopez-Mejia IC, Quiros PM, Bartoli C, Rivera J, Tazi J, Guzman G, Varela I *et al* (2011) Splicing-directed therapy in a new mouse model of human accelerated aging. *Sci Transl Med* 3: 106ra107

Vidak S, Serebryanny LA, Pegoraro G, Misteli T (2023) Activation of endoplasmic reticulum stress in premature aging via the inner nuclear membrane protein SUN2. *Cell Rep* 42: 112534

November 6, 2025

RE: Life Science Alliance Manuscript #LSA-2025-03485-TR

Dr. Roland Foisner
Medical University of Vienna
Max Perutz Labs
Dr. Bohr-Gasse 9
Vienna 1030
Austria

Dear Dr. Foisner,

Thank you for submitting your revised manuscript entitled "The unfolded protein response in progeria arteries originates from non-endothelial cell types". We have evaluated this revision and your rebuttal document internally without further reviewer input.

Overall we appreciate the constructive responses to all reviewer points, especially those we highlighted were required for consideration at LSA. We note the observation, made by the reviewers and editors, on the limits of in vitro culture that preclude certain assays on primary endothelial cells from aged mice. We concur the corroborating evidence provided using heart and aorta tissues, as well as RNA seq analysis, nicely addresses this concern. This relates, in part, to our second point on lung endothelial cells. Finally, we appreciate the addition of the Western blot in Fig 7D. In light of these improvements and the overall positive reception of this work by the original reviewers, we would be happy to publish your paper in Life Science Alliance pending final revisions necessary to meet our formatting guidelines.

- Please add a concluding statement to the abstract (and if you wish reducing the new sentence on scRNA seq analysis) similar to the statement at the close of the introduction.
- Please add an ORCID ID for the secondary corresponding author - they should have received instructions on how to do so.
- Please add the X and Bluesky handles of your host institute/organization, as well as your own and/or one of the authors, in our system.

A. FINAL FILES:

B. MANUSCRIPT ORGANIZATION AND FORMATTING:

Thank you for your attention to these final processing requirements. Please revise and format the manuscript and upload materials as soon as you are able.

Sincerely,

November 17, 2025

RE: Life Science Alliance Manuscript #LSA-2025-03485-TRR

Dr. Roland Foisner
Medical University of Vienna
Max Perutz Labs
Dr. Bohr-Gasse 9
Vienna 1030
Austria

Dear Dr. Foisner,

Thank you for submitting your Research Article entitled "The unfolded protein response in progeria arteries originates from non-endothelial cell types". It is a pleasure to let you know that your manuscript is now accepted for publication in Life Science Alliance. Thank you for resolving the remaining minor issues, including modification to the abstract, and congratulations on this interesting work.

DISTRIBUTION OF MATERIALS:

Again, congratulations on a very nice paper. I hope you found the review process to be constructive and are pleased with how the manuscript was handled editorially. We look forward to future exciting submissions from your lab.

Sincerely,
